# Online Iterative Reinforcement Learning from Human Feedback with General Preference Model

**Chenlu Ye**[*†]  **Wei Xiong**[* ‡]  **Yuheng Zhang**[*§]  **Hanze Dong**[*¶]

**Nan Jiang**[‖]  **Tong Zhang**[**]

## Abstract

We investigate Reinforcement Learning from Human Feedback (RLHF) in the context of a general preference oracle. In particular, we do not assume the existence of a reward function and an oracle preference signal drawn from the Bradley-Terry model as most of the prior works do. We consider a standard mathematical formulation, the reverse-KL regularized minimax game between two LLMs for RLHF under general preference oracle. The learning objective of this formulation is to find a policy so that it is consistently preferred by the KL-regularized preference oracle over any competing LLMs. We show that this framework is strictly more general than the reward-based one, and propose sample-efficient algorithms for both the offline learning from a pre-collected preference dataset and online learning where we can query the preference oracle along the way of training. Empirical studies verify the effectiveness of the proposed framework.

## 1  Introduction

*Reinforcement Learning from Human Feedback* (RLHF) has emerged as a pivotal technique in adapting machine learning to leverage relative feedback, especially in aligning Large Language Models (LLMs) with human values and preferences [14, 90]. Notable examples include ChatGPT [49], Claude [2], and Bard [29]. The primary goal of RLHF in the context of LLMs is to adjust the responses generated by LLMs so that they are more favorably received by human evaluators.

Inspired by the standard LLM alignment workflow [50, 5, 60], we characterize an LLM by a policy $\pi$, which takes a prompt $x \in \mathcal{X}$ and produces a response $a \in \mathcal{A}$ from the distribution $\pi(\cdot|x)$. In a typical LLM training pipeline [50, 60, 49], the tuning process begins with a pretrained model, which is subsequently fine-tuned using specialized and instructional data to produce an initial model $\pi_0$. The initial model $\pi_0$ is then aligned with a prompt set from some distribution $x \sim d_0$. The key component in RLHF is the *General Preference Oracle*, which is mathematically defined as follows.

**Definition 1** (General Preference Oracle)**.** *There exists a preference oracle $\mathbb{P} : \mathcal{X} \times \mathcal{A} \times \mathcal{A} \to [0, 1]$, and we can query it to receive the preference signal:*

$$y \sim \mathrm{Ber}\big(\mathbb{P}(a^1 \succ a^2|x, a^1, a^2)\big)$$

*where $y = 1$ means $a^1$ is preferred to $a^2$, and $y = 0$ means that $a^2$ is preferred.*

---

[*]Equal contributions with random author order. Correspondence to Wei Xiong.

[†]University of Illinois Urbana-Champaign. Email: chenluy3@illinois.edu

[‡]University of Illinois Urbana-Champaign. Email: wx13@illinois.edu

[§]University of Illinois Urbana-Champaign. Email: yuhengz2@illinois.edu

[¶]Salesforce AI Research. Email: hanze.dong@salesforce.com

[‖]University of Illinois Urbana-Champaign. Email: nanjiang@illinois.edu

[**]University of Illinois Urbana-Champaign. Email: tongzhang@tongzhang-ml.org

38th Conference on Neural Information Processing Systems (NeurIPS 2024).

Instead of directly optimizing against the preference oracle $\mathbb{P}$, the existing prevalent RLHF framework is reward-based [50, 60], which consists of three steps: (1) preference data collection, (2) reward modeling, and (3) policy optimization. Specifically, the preference dataset $\mathcal{D}$ consists of multiple tuples of the form $(x, a^1, a^2, y)$, whose collection process can be modeled as:

$$x \sim d_0, a^1 \sim \pi_D^1, a^2 \sim \pi_D^2, \qquad y \sim \text{Ber}\big(\mathbb{P}(a^1 \succ a^2 | x, a^1, a^2)\big), \qquad (1)$$

where $\pi_D^1$ and $\pi_D^2$ are behavior policies and are typically set as $\pi_0$ [60, 43] or some powerful closed-form LLMs [16]. The second step is reward modeling, which is the origin of the name "reward-based". This step can be viewed as a kind of inverse RL [89], which models some difficult-to-specify goals (preferred by the human or AI evaluators) as a scalar reward signal. Specifically, the Bradley-Terry (BT) model [9], a framework widely adopted in Ouyang et al. [50], Bai et al. [4], Touvron et al. [60], Rafailov et al. [53], Xiong et al. [72], assumes that there exists a ground-truth reward function $P^*$ and the preference model satisfies:

$$\mathbb{P}(a^1 \succ a^2 | x, a^1, a^2) = \frac{\exp(R^*(x, a^1))}{\exp(R^*(x, a^1)) + \exp(R^*(x, a^2))} = \sigma\big(R^*(x, a^1) - R^*(x, a^2)\big), \qquad (2)$$

where $\sigma(z) = 1/(1 + \exp(-z))$ is the sigmoid function. Then, the reward model is taken as the Maximum Likelihood Estimation (MLE) of the BT model on the preference dataset $\mathcal{D}$ [e.g., 51, 48, 50, 4, 60] and is used in subsequent policy optimization steps to provide a signal for algorithms like Proximal Policy Optimization [56]. Despite its successes, the existence of a reward function and the BT model are strong assumptions, which may not fully capture the complicated human preferences. In particular, the BT model assumes that human preference is transitive, which means that if we prefer A to B ($\mathbb{P}(A \succ B | x, A, B) > 0.5$) and we prefer B to C, then it automatically holds that we prefer A to C. This assumption, however, is contradicted by evidence of intransitivity in human decision-making [62, 45]. This limitation is particularly pronounced if we consider the population-level preferences, where the ultimate preference signal is aggregated across diverse human groups [45]. This may further be evidenced that in the practical RLHF, the accuracy of the learned BT model is around $70\%$ [4, 60, 16], suggesting the challenges in approximating the complicated human preference by BT model. While there are some recent efforts to bypass reward modeling [53, 85], they are still fundamentally derived from the reward-based preference model and suffer from the aforementioned issues. In contrast, the general preference oracle defined in Definition 1

Table 1: Comparison of the test accuracy between the BT-based reward model and the preference model. The reward model and preference model are trained with the same base model and preference dataset, where the details are deferred to Section 5. We evaluate the model on Reward-Bench [39].

| Base Model | Method | Chat | Chat Hard | Safety | Reasoning |
|---|---|---|---|---|---|
| Gemma-2B-it | BT | 95.0 | 40.8 | 81.2 | 74.2 |
| Gemma-2B-it | Preference | 96.0 | 40.5 | 82.8 | 80.7 |
| LLaMA3-8B-it | BT | 99.4 | 65.0 | 87.7 | 87.8 |
| LLaMA3-8B-it | Preference | 98.9 | 65.2 | 89.5 | 94.8 |

is strictly more general than the BT model and can capture a more complicated preference pattern from the definition itself. It allows an intransitive preference model and can further capture the preference feedback from AI [5], with a notable example of GPT-4 [49], which is widely used for model evaluations in practice and may more accurately reflect real user experience [60, 18, 53, 72]. Moreover, from a practical side, the preference model construction tends to be more efficient than the reward function in terms of ranking accuracy. This is evidenced by the fact that the preference model, pairRM with 0.4B parameters [34], performs comparably to a LLaMA2-13B-based reward model across a diverse set of preference targets [16]. As a case study, we train a reward model based on the Bradley-Terry (BT) model and a preference model with the same starting checkpoint Gemma-2B-it [59] and preference dataset[8], with results presented in Table 1 and the training details are deferred to Section 5. As we can see, the preference model achieves much higher test accuracy in the reasoning task while maintaining comparable results in other tasks. Meanwhile, the training set we use is rather limited in the reasoning data (math and coding), so the reasoning task can be

---

[8]We remark that the mixture of the open-source preference dataset and hyper-parameters are mainly tuned for the BT model with $> 2000$ A100 hours, while the preference model adopts most of them directly. Therefore, we expect that the preference model maybe even better with a more refined hyper-parameter search.

viewed as an out-of-distribution task. In this sense, the preference model may also provide a better generalization compared to the reward model. The results also extend to another case study with LLaMA3-8B-instruct, where the preference model shows promising potential in the improvement of reasoning tasks. We refer interested readers to check Zhao et al. [85], Liu et al. [43] for further examples with similar observations. The advantage in ranking accuracy is not only directly beneficial for the algorithms that depend on ranking information [18, 30], but also improves the performance of algorithms derived from the reward-based framework (i.e., Bradley-Terry model), as evidenced by the results in the study of (iterative) DPO [72, 31].

Given all these considerations, our study focuses on exploring the theoretical properties of RLHF under the general preference oracle (Definition 1), with the goal of advancing practical algorithmic designs. We summarize our contributions as follows:

- We make the first attempt to study the theoretical learnability of RLHF under general preference oracle with KL regularization, in the offline setting with a pre-collected preference dataset and the online setting where we can query human feedback along the way of training, which demonstrates the potential of reward-model-free learning under general preference;
- We propose sample-efficient algorithms in both the offline setting and online setting and establish the finite-sample theoretical guarantees under standard coverage and exploration conditions;
- We show that the theoretical insights can be used to guide practical algorithmic designs with a reasonable approximation of the computational oracle.

## 2 Problem Formulation

In this section, we formulate the RLHF with general preference learning. Suppose that there exists a preference function $P^* : \mathcal{X} \times \mathcal{A} \times \mathcal{A} \to \mathbb{R}$ which represents the prefererence of one action $a^1$ over another $a^2$ given a prompt $x$: $P^*(x, a^1, a^2) = \mathbb{P}(a^1 \succ a^2 | x, a^1, a^2)$. In practical applications, we want to make the resulting LLM $\pi$ close to $\pi_0$ [90, 50, 4, 53]. Therefore, we adopt the following KL-regularized objective:

$$J(\pi^1, \pi^2) = \mathbb{E}_{x \sim d_0} \mathbb{E}_{a^1 \sim \pi^1, a^2 \sim \pi^2} \Big[ P^*(x, a^1, a^2) - \eta^{-1} D_{\mathrm{KL}}(\pi^1(\cdot|x) \| \pi_0(\cdot|x)) + \eta^{-1} D_{\mathrm{KL}}(\pi^2(\cdot|x) \| \pi_0(\cdot|x)) \Big].$$
(3)

One primary reason to consider the regularized target is that the constructed preference model is only *locally* accurate, i.e., it performs well when there is little distribution shift. For instance, if the preference model is fine-tuned on a preference dataset collected by the initial model $\pi_0$, it improves the in-distribution generalization, but the resulting model often performs poorly out-of-distribution [10]. Meanwhile, even if we require human labelers to give feedback along the way, the choices of the labelers may not be representative enough or the labelers can make mistakes due to limited time, attention, or care [27]. Moreover, the KL divergence in the target ensures that the resulting policy is stochastic instead of deterministic (given a suitable initial checkpoint), thereby more accurately reflecting the dynamics of generative language models.

We choose $P^*$ as the target mostly for historical reasons [22, 65]. A choice is the relative preference $\log(P^*(x, a^1, a^2)/(1 - P^*(x, a^1, a^2)))$, which is equal to $R^*(x, a^1) - R^*(x, a^2)$ when the BT model holds so that (3) becomes two decoupled regularized-reward maximization problems in this case and automatically reduces to the setting considered in the previous work Xiong et al. [72]. While we do not handle this target directly, the analysis techniques presented in this paper readily apply to it with slight modifications.

**Nash Equilibrium and Best Response.** Without loss of generality, we restrict our attention to the policy class $\Pi$ consisting of the policies with the same support as $\pi_0$ and denote the *unique* Nash equilibrium (known as the Minimax Winner [57, 38, 24] or the von Neumann Winner [22]) as the solution of the following minimax problem as:

$$(\pi^1_*, \pi^2_*) = (\pi_*, \pi_*) = \underset{\pi^1 \in \Pi}{\mathrm{argmax}} \, \underset{\pi^2 \in \Pi}{\mathrm{argmin}} \, J(\pi^1, \pi^2),$$
(4)

where the Nash policies of two players coincide as we prove in Lemma 4. In the rest of this paper, we still use the notation $(\pi^1_*, \pi^2_*)$ to distinguish between the max-player and min-player. Accordingly, we refer to the first LLM $\pi^1$ as the *max-player*, while the second LLM $\pi^2$ is the *min-player*. We also

define the notion of *best response*. For function $J$ and policy $\pi^1$, the best response to $\pi^1$ is defined as $\arg\min_{\pi^2 \in \Pi} J(\pi^1, \pi^2)$ and the value is denoted by $J(\pi^1, \dagger) = \min_{\pi^2 \in \Pi} J(\pi^1, \pi^2)$. Similarly, for $\pi^2$, we have $J(\dagger, \pi^2) = \max_{\pi^1 \in \Pi} J(\pi^1, \pi^2)$. In particular, since $\pi_*^1$ and $\pi_*^2$ are the Nash equilibrium, they are the best response to each other.

**Function Approximation.** Suppose that we have access to a function class $\mathcal{P} \subset (\mathcal{X} \times \mathcal{A} \times \mathcal{A} \to \mathbb{R})$ (e.g. neural network), which provides us with a set of candidates to approximate the $P^*$, and also the preference functions $P \in \mathcal{P}$ satisfies $P(x, a^1, a^2) = 1 - P(x, a^2, a^1)$. We make the following assumptions on the class $\mathcal{P}$.

**Assumption 1.** *Assume that $\mathcal{P}$ is finite and the capacity of the class is large enough so that $P^* \in \mathcal{P}$.*

The finite class assumption is for a clear presentation and the results readily generalize to an infinite class with a bounded covering number by the standard discretization technique. We define a theoretical computation oracle as follows and defer the practical implementations to the experiment section.

**Definition 2** (Nash Equilibrium Oracle). *For a given preference function $P \in \mathcal{P}$ and a reference policy $\pi_0$, we can compute the Nash Equilibrium policy*

$$\pi_P = \underset{\pi^1 \in \Pi}{\arg\max} \min_{\pi^2 \in \Pi} \mathbb{E}_{x \sim d_0} \mathbb{E}_{a^1 \sim \pi^1, a^2 \sim \pi^2} \left[ P(x, a^1, a^2) - \eta^{-1} \log \frac{\pi^1(a^1|x)}{\pi_0(a^1|x)} + \eta^{-1} \log \frac{\pi^2(a^2|x)}{\pi_0(a^2|x)} \right]. \quad (5)$$

**Learning Objective.** The goal is to find an $\epsilon$-approximate Nash policy $\hat{\pi}^1$ for the max-player:

$$J(\pi_*^1, \pi_*^2) - J(\hat{\pi}^1, \dagger) = J(\pi_*^1, \pi_*^2) - \min_{\pi'} J(\hat{\pi}^1, \pi') \leq \epsilon,$$

which means that the max-player is consistently preferred by the KL-regularized preference in the face of any competing policy $\pi'$ up to a relaxation of $\epsilon$. To stress the non-symmetric structures of the two players, we refer to the max-player as the *main agent*, which aims to find her $\epsilon$-approximate Nash policy, and refer to the min-player as the *enhancer*, which is designed to facilitate the main agent's learning. In particular, when $\eta$ is large enough so that the KL is roughly omitted, then, we can further obtain that

$$\min_{\pi^2 \in \Pi} \mathbb{E}_{x \sim d_0} \mathbb{E}_{a^1 \sim \hat{\pi}^1, a^2 \sim \pi^2} P^*(x, a^1, a^2) \geq 0.5 - \epsilon.$$

In this case, the obtained policy $\hat{\pi}_1$ is consistently preferred by the preference oracle $P^*$ against any competing policies. We mention in passing that the KL penalty coefficient $\eta > 0$ exhibits a trade-off between being preferred by the oracle $P^*$ and staying close to the initial model $\pi_0$, and reflects the degree of our belief in the oracle $P^*$. In practice, $\eta$ is typically treated as a hyper-parameter and is adjusted by parameter search [32].

Compared to the previous literature formulating the preference learning as finding a Nash equilibrium, although we focus on optimizing the policy for the max-player, we can also have a duality gap guarantee because of the symmetry of the objective function: $J(\pi^1, \pi^2) = 1 - J(\pi^2, \pi^1)$. To see this, we decompose the duality gap into the suboptimality for the max-player $\hat{\pi}^1$ and the min-player $\hat{\pi}^2$:

$$J(\dagger, \hat{\pi}^2) - J(\hat{\pi}^1, \dagger) = J(\dagger, \hat{\pi}^2) - J(\pi_*^1, \pi_*^2) + J(\pi_*^1, \pi_*^2) - J(\hat{\pi}^1, \dagger)$$
$$= J(\pi_*^1, \pi_*^2) - J(\hat{\pi}^2, \dagger) + J(\pi_*^1, \pi_*^2) - J(\hat{\pi}^2, \dagger).$$

If we obtain such an $\epsilon$-suboptimal max player $\hat{\pi}^1$, by taking the min-player $\hat{\pi}^2 = \hat{\pi}^1$, the duality gap $J(\dagger, \hat{\pi}^2) - J(\hat{\pi}^1, \dagger)$ is naturally bounded by $2\epsilon$.

**Notations.** We use the short-hand notation $\pi = (\pi^1, \pi^2)$ when there is no confusion. We use $P(x, \pi^1, \pi^2)$ to represent $\mathbb{E}_{a^1 \sim \pi^1, a^2 \sim \pi^2}[P(x, a^1, a^2)]$. We use $J(x, \pi^1, \pi^2)$ to denote the objective function in (3) without the expectation over the prompt $x \sim d_0$. Let $\sigma(x)$ denote the sigmoid function $1/(1 + e^{-x})$. We also provide a notation table in Table 4 to improve the readability of this paper.

Due to space constraints, the review of the related literature is deferred to Appendix 7.

## 3 Improved Algorithms in Offline Setting

### 3.1 Setup

In the offline setting, our goal is to learn a good policy from a pre-collected dataset $\mathcal{D}_{\text{off}} = \{(x_i, a_i^1, a_i^2, y_i)\}_{i=1}^n$ without further query with the oracle $\mathbb{P}$, where comparison sample is assumed

---

**Algorithm 1** Pessimistic Equilibrium Learning from Human Feedback

---

1: **Input:** Dataset $\mathcal{D}_{\text{off}} = \{x_i, a_i^1, a_i^2, y_i\}_{i=1}^n$, preference space $\mathcal{P}$, policy class $\Pi$, parameter $\eta, \beta > 0$.

2: Compute the MLE $\hat{P} = \operatorname{argmin}_{P \in \mathcal{P}} \ell_{\mathcal{D}_{\text{off}}}(P)$.

3: Construct version space

$$\widehat{\mathcal{P}} = \Big\{ P \in \mathcal{P} : \sum_{i=1}^n (P(x_i, a_i^1, a_i^2) - \hat{P}(x_i, a_i^1, a_i^2))^2 \le \beta^2/2 \Big\}. \tag{8}$$

4: Compute the best policy under the conservative value estimation

$$\hat{\pi}^1 = \operatorname*{argmax}_{\pi^1 \in \Pi} \min_{\pi^2 \in \Pi} \min_{P \in \widehat{\mathcal{P}}} \mathbb{E}_{x \sim d_0} \mathbb{E}_{a^1 \sim \pi^1, a^2 \sim \pi^2} \Big[ P(x, a^1, a^2) + \eta^{-1} \ln \frac{\pi_0(a^1|x)}{\pi^1(a^1|x)} - \eta^{-1} \ln \frac{\pi_0(a^2|x)}{\pi^2(a^2|x)} \Big]. \tag{9}$$

5: **Output:** $\hat{\pi}^1$.

---

to be independently collected as in (1). We measure the suboptimality of the learned policy $\hat{\pi}^1$ by the gap between the Nash value and the best response value:

$$J(\pi_1^*, \pi_2^*) - J(\hat{\pi}^1, \dagger), \tag{6}$$

where the KL-regularized function $J$ is defined in (3). Similar to the reward-based framework [50], one natural approach is a two-staged method: (1) Construct an empirical preference model (reward model in the literature) by maximizing the log-likelihood function:

$$\ell_{\mathcal{D}_{\text{off}}}(P) = \sum_{(x, a^1, a^2, y) \in \mathcal{D}_{\text{off}}} y \log P(x, a^1, a^2) + (1 - y) \log P(x, a^2, a^1); \tag{7}$$

(2) Solve the policy by plugging the learned preference model $\hat{P}$ into the Nash Equilibrium Oracle 2. However, this framework typically leads to severe reward over-optimization issue [26], meaning that while the model is preferred by the learned $\hat{P}$, it may not achieve good performance under the evaluation of $P^*$. This is because, with finite $\mathcal{D}_{\text{off}}$ drawn from some behavior policy, it is unlikely to provide an accurate estimation for all the prompt-response pairs. Therefore, imposing heavy optimization pressure toward $\hat{P}$ will push the model to exploit these unreliable estimations to chase for a high proxy metric, thus leading to a worse performance under the ground truth $P^*$.

### 3.2 Learning with Pessimism

The recent advances in the offline RL theory have demonstrated that the principle of pessimism with a conservative estimation is statistically efficient for offline learning across a diverse set of scenarios [35, 54, 69, 79, 86, 17, 71, 84]. In this section, we connect the KL-reversed minimax game in (3) with offline RL by pessimism via version space[9].

We introduce our algorithm, Pessimistic Equilibrium Learning from Human Feedback (PELHF) in Algorithm 1. Given an offline dataset $\mathcal{D}_{\text{off}}$, we first obtain the maximum likelihood estimation (MLE) $\hat{P}$ by maximizing (7). Rather than directly planning with this empirical $\hat{P}$, we form a version space $\widehat{\mathcal{P}}$ that contains $P^* \in \widehat{\mathcal{P}}$ with a high probability under a suitable choice of $\beta$, as we show in Lemma 1. For each policy $\pi^1$, we take the minimum preference function over $\widehat{\mathcal{P}}$ and the best responded $\pi^2$ as its conservative value estimation:

$$\hat{J}_{\text{off}}(\pi^1) = \min_{\pi^2 \in \Pi} \min_{P \in \hat{P}} \mathbb{E}_{x \sim d_0} \mathbb{E}_{a^1 \sim \pi^1, a^2 \sim \pi^2} \Big[ P(x, a^1, a^2) + \eta^{-1} \ln \frac{\pi_0(a^1|x)}{\pi^1(a^1|x)} - \eta^{-1} \ln \frac{\pi_0(a^2|x)}{\pi^2(a^2|x)} \Big].$$

Then, we solve the minimax game concerning this conservative value estimator. With this pessimistic modification, the resulting algorithm enjoys the following theoretical guarantee.

**Theorem 1.** *[Proof] If Assumption 1 holds, and we set $\lambda = \log(|\mathcal{P}|/\delta)$ and $\beta^2 = 2\log(|\mathcal{P}|/\delta)$, then, with probability at least $1 - \delta$, the output policy of Algorithm 1 satisfies*

$$J(\pi_1^*, \pi_2^*) - J(\hat{\pi}^1, \dagger) \le 4\beta \sqrt{\mathcal{C}(\pi_*^1, \pi_D, \mathcal{P})/n}.$$

*where the coverage coefficient*

$$\mathcal{C}(\pi_*^1, \pi_D, \mathcal{P}) = \max_{\pi^2 \in \Pi} \sup_{P \in \mathcal{P}} \frac{(\mathbb{E}_{x \sim d_0}[P(x, \pi_*^1, \pi^2) - \hat{P}(x, \pi_*^1, \pi^2)])^2}{\mathbb{E}_{x \sim d_0, a^1 \sim \pi_D^1, a^2 \sim \pi_D^2}(P(x, a^1, a^2) - \hat{P}(x, a^1, a^2))^2}.$$

---

[9]We introduce another algorithm achieving pessimism via uncertainty bonus construction, see Appendix C.2.

This theorem shows that the suboptimality gap depends on how the target $(\pi_*^1, \pi^2)$ is covered by the offline dataset, where $\pi^2$ is maximized over the policy set $\Pi$. This coverage coefficient resembles the unilateral coverage[10] for Markov games [17, 86]. Then, a natural question is whether a good coverage condition ($\mathcal{C}(\pi_*^1, \pi_D, \mathcal{P})$ is small) is practical in the context of LLMs. Unfortunately, since the response is usually long in practice, the distribution shift between policies is also very large. We summarize some observations here. First, along the way of the RLHF training, the average density ratio $\pi(a|x)/\pi_0(a|x) > \exp(25)$ as reported in Figure 13 of Bai et al. [4]. See similar results of rejection sampling fine-tuning [18] and DPO [53]. Second, for a case study, we use the Gemma-7B-it as the behavior policy to collect data for aligning Gemma-2B-it [59] with 15k prompt from [16]. Then, we calculate the average KL divergence between Gemma-7B-it and Gemma-2B-it as $456.4$. This evidence indicates that the coverage coefficient probably explodes in realistic scenarios. Therefore, it is unlikely to expect that we can learn the optimal policy from a pre-collected dataset. This motivates us to consider the online setting, where we can further query the preference oracle during the training to enrich the dataset thus enhancing our models continuously.

# 4 Iterative RLHF with Online Exploration

## 4.1 Setup of Iterative RLHF

The major difference between the online and offline settings is that online algorithms can further query the preference oracle $P^*$ along the way of training. Since updating the LLMs is expensive, we consider the batch online setting for a sparse policy update. Specifically, for each batch $t \in [T]$, we first update the policy pair $(\hat{\pi}_t^1, \hat{\pi}_t^2)$ based on the historical information collected so far. Then, we collect $m$ tuples: we sample a random prompt by $x_{t,i} \sim d_0$, collect two responses by $(a_{t,i}^1, a_{t,i}^2) \sim (\hat{\pi}_t^1, \hat{\pi}_t^2)$, and query the preference signal $y_{t,i} \sim \text{Ber}(P^*(x_{t,i}, a_{t,i}^1, a_{t,i}^2))$. Here the batch size $m$ is usually very large compared to the typically adopted mini-batch update. To distinguish this from the sequential online setting where we update policy after collecting a single preference pair, we refer to this learning paradigm as the *iterative RLHF*.

## 4.2 Learning with Exploration

The primary advantage of online learning is that we can strategically choose the behavior policies in each iteration to improve the coverage of the collected data, which is referred to as the exploration in the literature. To achieve this goal, we need to quantify the data uncertainty to guide the exploration direction. To this end, we present the notions of information ratio and eluder coefficient.

**Information Ratio and Eluder Coefficient.** Distinct from the offline setting where we assume the coverage condition of a pre-collected dataset $\mathcal{D}_{\text{off}}$, online exploration makes it possible to upper bound the suboptimality by the complexity of the function space. We leverage the notion of the eluder coefficient, which limits the generalization from visited state-action distributions to unseen parts.

**Definition 3** (Information Ratio and Eluder Coefficient). *At round t, given an estimation $\hat{P} \in \mathcal{P}$, we define the information ratio for any two policy $\pi^1, \pi^2$ as*

$$\Gamma_t(\lambda, \pi^1, \pi^2) = \sup_{P \in \mathcal{P}} \frac{|\mathbb{E}_{x \sim d_0}[P(x, \pi^1, \pi^2) - \hat{P}(x, \pi^1, \pi^2)]|}{\sqrt{\lambda + \sum_{s=1}^{t-1} \mathbb{E}_{x_s \sim d_0, a_s^1 \sim \hat{\pi}_s^1, a_s^2 \sim \hat{\pi}_s^2}(P(x_s, a_s^1, a_s^2) - \hat{P}(x_s, a_s^1, a_s^2))^2}}.$$

*Then, the eluder coefficient is given by $d(\mathcal{P}, \lambda, T) := \sup_{\pi_{1:T}^1, \pi_{1:T}^2} \sum_{t=1}^{T} \min(1, (\Gamma_t(\lambda, \pi_t^1, \pi_t^2))^2)$.*

The information ratio and eluder coefficient considered here have also been adopted in the literature [e.g., 64, 28, 70, 74, 1]. Essentially, the information ratio compares the *out-of-sample* error on the unseen data with the *in-sample* error measured on the historical data, and can be interpreted as the worst-case ratio between them (as we take supreme over all possible $P \in \mathcal{P}$). Meanwhile, the eluder coefficient limits the extent to which we can be "surprised" by the new out-of-sample distributions, given the historical data collected so far. The uncertainty for the preference model aligns with the uncertainty for the BT model under boundedness conditions, which is illustrated in the following example. We defer the details to Appendix D.1.

---

[10]In Appendix C.3, we show that with an improved analysis, Algorithm 1 enjoys a refined coverage condition, similar to the coverage notion in [84].

**Example 1** (Uncertainty in Bradley-Terry model with linear reward). *Suppose the reward function can be embedded into a $d$-dimensional vector space $\{r(x,a) = \langle \theta, \phi(x,a) \rangle : \theta \in \mathbb{R}^d, \|\theta\| \leq B, \|\phi(x,a)\| \leq 1\}$. Then, if we define the covariance matrix as $\Sigma_t = \sum_{s=1}^{t-1} \mathbb{E}_{x \sim d_0, a^1 \sim \hat{\pi}_s^1, a^2 \sim \hat{\pi}_s^2} (\phi(x,a^1) - \phi(x,a^2))^\top (\phi(x,a^1) - \phi(x,a^2)) + \lambda(1+e^B)^2 I$, we have*

$$\Gamma_t(\lambda, \pi^1, \pi^2) \leq (1+e^B) \|\phi(x,\pi^1) - \phi(x,\pi^2)\|_{\Sigma_t^{-1}}.$$

---

**Algorithm 2** Optimistic Equilibrium Learning from Human Feedback with Enhancer

1: **Input:** Preference space $\mathcal{P}$, policy class $\Pi$, parameter $\eta, \lambda > 0$.
2: **for** t=1,...,T **do**
3:    Exploitation with the main agent: compute the MLE $\hat{P}_t$ with $\ell_{\mathcal{D}_{1:t-1}}$ defined in (7) and compute Nash equilibrium by calling the Nash equilibrium oracle 2:
$$\hat{\pi}_t^1 = \underset{\pi^1 \in \Pi}{\arg\max} \min_{\pi^2 \in \Pi} \mathbb{E}_{x \sim d_0, a^1 \sim \pi^1, a^2 \sim \pi^2} \left[ \hat{P}_t(x, a^1, a^2) + \eta^{-1} \log \frac{\pi_0(a^1|x)}{\pi^1(a^1|x)} - \eta^{-1} \log \frac{\pi_0(a^2|x)}{\pi^2(a^2|x)} \right],$$
(10)
4:    Exploration with the enhancer: compute enhancer to maximize the uncertainty:
$$\pi_t^2 = \underset{\pi^2 \in \Pi}{\arg\max} \widetilde{\Gamma}_t^m(\lambda, \hat{\pi}_t^1, \pi^2) := \sup_{P \in \mathcal{P}} \frac{|\mathbb{E}_{x \sim d_0}[P(x, \hat{\pi}_t^1, \pi^2) - \hat{P}_t(x, \hat{\pi}_t^1, \pi^2)]|}{\sqrt{\lambda + \frac{1}{m} \sum_{s=1}^{t-1} \sum_{j=1}^m (P(x_{s,j}, a_{s,j}^1, a_{s,j}^2) - \hat{P}_t(x_{s,j}, a_{s,j}^1, a_{s,j}^2))^2}},$$
(11)
5:    Collect $\mathcal{D}_t = \{(x_i, a_i^1, a_i^2, y_i)\}_{i=1}^m$ by $x_i \sim d_0, a_i^1 \sim \hat{\pi}_t^1(\cdot|x_i), a_i^2 \sim \hat{\pi}_t^2(\cdot|x_i)$ and $y_i \sim \mathrm{Ber}(\mathbb{P}(a_i^1 \succ a_i^2|x, a_i^1, a_i^2))$;
6: **end for**
7: **Output:** the best policy in $(\pi_{1:T}^1)$ by a validation set.

---

We refer interested readers to Du et al. [20], Zhong et al. [87], Xie et al. [70] for the extensive examples when $d(\mathcal{P}, \lambda, T)$ can have a sub-linear dependency on $T$. We are now ready to present the algorithm for the online setting, as summarized in Algorithm 2. Specifically, for each iteration, the main agent exploits the information contained in the data collected so far by computing the MLE $\hat{P}_t$ and solving the minimax game with respect to it to get $\hat{\pi}_t^1$. The enhancer, however, aims to facilitate the main agent's learning by maximizing the uncertainty relative to the $\hat{\pi}_t^1$. Finally, we use the policy pair to collect $m$ preference pairs and query oracle $P^*$ to get the preference signals. Notably, to facilitate the computation for the main agent, instead of adding optimism to the value function, we impose the exploration role on the enhancer. This choice turns out to be important when we move toward practical algorithms with reasonable approximations, as we detail in Section 5. We now present the main theoretical guarantee for Algorithm 2.

**Theorem 2.** *[Proof] Under Assumption 1, for any $\epsilon > 0$, if we set the total iterations $T = \min\{n \in \mathbb{N}^+ : n \geq 2d(\mathcal{P}, \lambda, n)\}$, batch size $m = 18T \log(2T|\mathcal{P}|/\delta)/\epsilon^2$, $\beta = \sqrt{2T \log(2T|\mathcal{P}|/\delta)/m}$, and $\lambda = 2T \log(2T|\mathcal{P}|/\delta)/m$ for Algorithm 2, then, with probability at least $1-\delta$, there exists a $t_0 \in [T]$,*

$$J(\pi_*^1, \pi_*^2) - J(\hat{\pi}_{t_0}^1, \dagger) \leq \epsilon.$$

The theorem states that with suitable hyper-parameter choices, after $T$ iterations (up to log factors), we can find an $\epsilon$-approximate Nash policy $\hat{\pi}_{t_0}^1$ for the max-player. Here $T$ depends on the eluder coefficient that is intrinsic to the preference model and characterizes the complexity of the problem.

**Key Ideas.** We present a brief discussion of the key analysis ideas. Similar to Lemma 1, the MLE $\hat{P}$ ensures a controllable in-sample error (with details in the Appendix D). Recalling that the uncertainty bonus is essentially the worst-case ratio between the out-of-sample error (our learning target) and the in-sample error, to finally bound the out-of-sample error, we need to explore each direction where we are uncertain about so that the average uncertainty bonus is sufficiently small. Since the main agent is greedy (takes the best guess we can obtain so far), the enhancer plays the exploration role by maximizing the uncertainty relative to the $\hat{\pi}_t^1$. Then, since the eluder dimension is finite: $\sum_{t=1}^T \min\left(1, (\Gamma_t(\lambda, \hat{\pi}_t^1, \hat{\pi}_t^2))^2\right) \leq d(\mathcal{P}, \lambda, T)$, there exists at least a $t_0 \in [T]$ such that the value at $t_0$ is smaller or equal to the average value:

$$\min\left(1, (\Gamma_t(\lambda, \hat{\pi}_t^1, \hat{\pi}_t^2))^2\right) \leq d(\mathcal{P}, \lambda, T)/T \leq 1/2.$$

Hence, with a proper $m$, we can obtain the result of Theorem 2.

In practice, searching for the most uncertain policy in the whole policy space can be challenging and the enhancer policy itself does not enjoy any theoretical guarantee. We may slightly modify Algorithm 2 by restricting the exploration step to the following subset

$$\Pi_t = \{\pi \in \Pi : \eta^{-1}\mathbb{E}_{x \sim d_0}D_{\mathrm{KL}}(\pi(\cdot|x), \hat{\pi}^1(\cdot|x)) \leq \beta(\widetilde{\Gamma}_t^m(\lambda, \hat{\pi}^1, \pi) + \widetilde{\Gamma}_t^m(\lambda, \hat{\pi}^1, \hat{\pi}^1))\}, \quad (12)$$

where $\beta$ is the parameter defined in Theorem 2. This set is never empty because we can prove that both $\hat{\pi}_t^1$ and $\mathrm{argmin}_{\pi'} J(\hat{\pi}_t^1, \pi')$ belong to $\Pi_t$. Intuitively, maintaining a small KL divergence against $\hat{\pi}_t^1$ corresponds to exploiting the historical information, and maximizing the uncertainty relative to $\hat{\pi}_t^1$ leads to more information gain. The choice of $\Pi_t$ represents a refined trade-off between these two different goals, thus making $\hat{\pi}_t^2$ also converge to $\pi_*$. The details are deferred to Appendix D.2.

## 5 Practical Implementation of Preference Model and Iterative RLHF

In this section, we discuss how to implement the theoretical Algorithm 2 for the online setting.

**Main agent approximates Nash equilibrium oracle via self-play IPO.** Approximating the information-theoretical oracle 2 given a known preference model has been studied in Munos et al. [46], Calandriello et al. [11]. The proposed algorithm, self-play IPO, can serve as a reasonable approximation of the oracle by optimizing the following loss function:

$$\mathbb{E}_{x \sim d_0, a, a' \sim \mathrm{SG}[\pi], a^+, a^- \sim \hat{P}_t(x, a, a')}\left[\log \frac{\pi(a^+|x)\pi_0(a^-|x)}{\pi(a^-|x)\pi_0(a^+|x)} - \frac{1}{2\eta}\right]^2, \quad (13)$$

where $\mathrm{SG}[\pi]$ means that although we generate data from policy $\pi$, but we do not compute the gradient for this data-generation process. Moreover, according to Proposition 4.1 of Calandriello et al. [11], the minimizer of (13) is the unique Nash policy of the (10).

**Enhancer explores via rejection sampling.** According to (12), the enhancer aims to find a policy that (1) is close to the main agent's policy $\hat{\pi}_t^1$; (2) maximizes the uncertainty relative to the $\hat{\pi}_t^1$. However, since for the general neural network, the uncertainty estimator does not admit a closed form, in practice, we typically resort to heuristic methods. One popular way in the context of alignment is the *rejection sampling* [47, 18, 43, 31, 76]. Specifically, given a prompt $x$, we use $\hat{\pi}_t^1$ to independently sample $n$ responses, use a tournament-style procedure to get the best response (and reject all other responses), and take the best responses as $\hat{\pi}_t^2$. In other words, we take the policy induced by rejection sampling with $\hat{\pi}_t^1$ and $P^*$ as the enhancer policy $\hat{\pi}_t^2$. In this way, the $\hat{\pi}_t^2$ enlarges the margins between $\hat{\pi}_t^1$ while maintaining a moderate KL divergence. For instance, in the special case of the BT model, if we rank the samples via the learned reward, the KL divergence is upper bounded by $\log n - (n-1)/n$ and is usually far better than this conservative estimation [6].

**Preference model construction.** We follow Zhao et al. [85], Liu et al. [43], Dong et al. [19] to utilize the fact that the LLM is the next token predictor for the preference modeling. Specifically, we have a preference pair $(x, a^1, a^2, A)$, where $A$ means that the first response is better, which is formatted as

instruction = [CONTEXT] {x} [RESPONSE A] {$a^1$} [RESPONSE B] {$a^2$}, and label = A.

Then, we simply treat the preference modeling as an instruction-following task to fine-tune the model on these instruction-label pairs. In particular, to mitigate the position bias (the preference model may prefer the response that is given in the position of RESPONSE A), we randomly switch the order of the two responses in the data formatting process. During inference, we simply use the probability of decoding A as the $\hat{P}(x, a^1, a^2)$. We mention in passing that it is also possible to include a rubric in the instruction template to guide the model's prediction and achieve better results [52]. We observe the benefits of the additional prompt engineering in early experiments but decide to use the current version because the main focus is to verify the effectiveness of general preference structure. This implementation is also referred to as the **Generative RM** in subsequent works.

## 6 Experiments

**Model, Dataset, and Evaluation.** We adopt the widely used open-source model Zephyr-SFT-7B [61] as the starting checkpoint, which is based on the Mistral-7B-v0.1[11] and fine-tuned on 200K

---

[11]`https://huggingface.co/mistralai/Mistral-7B-Instruct-v0.1`

Table 2: The evaluation results of the IPO-aligned models under different KL coefficients. For the first 4 win rates, we use the LLaMA3-8B-based preference model to conduct head-to-head comparisons on the hand-out test set from Ultra-feedback with 3K prompts.

| MODELS | v.s. SFT | v.s. $\eta = 0.1$ | v.s. $\eta = 0.5$ | v.s. $\eta = 1.0$ | ALPACAEVAL2 |
|---|---|---|---|---|---|
| SFT | 0.5 | 0.121 | 0.205 | 0.231 | 4.63 |
| OFFLINE-IPO-$\eta = 0.1$ | **0.879** | **0.5** | **0.673** | **0.769** | **9.36** |
| OFFLINE-IPO-$\eta = 0.5$ | 0.795 | 0.327 | 0.5 | 0.632 | 6.86 |
| OFFLINE-IPO-$\eta = 1.0$ | 0.710 | 0.230 | 0.328 | 0.5 | 6.55 |

Table 3: The evaluation results of the models from different RLHF algorithms. The gold win rates are computed on the hand-out test set from Ultra-feedback with 3K prompts, with the Offline DPO model as the reference. Details of AlpacaEval2 can be found in Dubois et al. [21].

| MODELS | SETTINGS | GOLD WR v.s. IPO | ALPACAEVAL2 WR |
|---|---|---|---|
| SFT | - | 0.121 | 4.63 |
| OFFLINE DPO | OFFLINE | 0.41 | 9.33 |
| OFFLINE IPO | OFFLINE | 0.5 | 9.36 |
| ONLINE-ELHF-IPO | ONLINE | **0.78** | **17.67** |

high-quality Ultra-chat data [16]. We use the Ultra-feedback [16] as our prompt set. We divide the prompt set into the train set (60K), validation set (1K), and test set (3K). We mainly use head-to-head comparisons to evaluate the resulting models. In particular, we consider two types of win rate: 1) the win rate measured by the ground-truth LLaMA3-8B-based preference model on the hand-out test set from UltraFeedback; 2) the win rate measured by the GPT-4 Preview (11/06) on an out-of-distribution prompt set AlpacaEval2 [21]. Specifically, for the first evaluation, we use the best DPO model as the reference model, and for the AlpacaEval2, the GPT-4 Preview (11/06) is used as a reference model, and as the judge at the same time.

**Method and Competitors.** We consider the implementation of Algorithm 2 with self-play IPO and rejection sampling as discussed in Section 5. We iterate for three iterations in total and for each iteration, we retrain a preference model using all the historical data, and run self-play IPO from the initial checkpoint $\pi_0$ (i.e., Zephyr-7B-SFT). For simplicity, we refer to this algorithm as Online ELHF IPO. We use the offline DPO [53], offline IPO [3], and SFT model as the baseline. In particular, we do not further fine-tune the Zephyr-7B-SFT on the preferred responses of Ultra-Feedback because the quality of Ultra-Feedback is lower than that of Ultra-Chat, which is generated by Chat-GPT APIs. For DPO, we follow Xiong et al. [72], Tunstall et al. [61], Rafailov et al. [53] to set the KL coefficient as $\eta = 0.1$. For IPO, we search the hyper-parameter in $\{0.1, 0.5, 1.0\}$ and report the results in Table 2. Clearly, the model with $\eta = 0.1$ beats all other IPO models and the SFT model with large margins, so we set $\eta = 0.1$ for the offline IPO and the Online ELHF IPO algorithm in the subsequent studies.

**Simulation framework.** For all the offline algorithms, we sample two responses per prompt of the train set and use the LLaMA3-8B-based preference model to give the preference signal. Then, we run offline DPO and IPO with the synthetic dataset. For the Online ELHF, we set $n = 4$ in the rejection sampling process and use a tournament-style ranking method (so that the complexity of rejection sampling is linear in $n$) to find the best response.

**IPO, DPO, and Online ELHF-IPO.** We use the open-source project TRL[12] to implement IPO and DPO. In particular, we have implemented IPO with log-likelihood/perplexity (perplexity is averaged log-likelihood by sequence length), where the original authors of IPO suggest that log-likelihood-based implementation is unstable (see the huggingface blog[13] for details). We also found that the IPO without average cannot normally converge and is of poor performance and take the perplexity implementation accordingly. For DPO, we implement the vanilla version as the baseline. We present the main result in Table 3. It is clear that Online ELHF-IPO outperforms the baselines.

---

[12]https://github.com/huggingface/trl
[13]https://huggingface.co/blog/pref-tuning

# 7 Related Work

This section focuses on the theoretical aspects. A general discussion is provided in Appendix B.1

**Theoretical Study of Reward-based RLHF.** The theoretical study of policy optimization from preference feedback dated back to the dueling bandits [e.g., 78, 55, 7]. This was later extended to the online RL setting by Xu et al. [73], Novoseller et al. [48], Pacchiano et al. [51], Chen et al. [12], including tabular online RLHF with finite state, and general function approximation for capturing real problems with large state spaces. Zhan et al. [81], Wu and Sun [68] further encompasses the development of reward-free learning type algorithms and sampling-based algorithms for online RLHF. Apart from the online setting, there is another line of works [88, 80, 40] studying the reward-based RLHF in the offline setting, which learns from a pre-determined offline dataset with suitable coverage condition over the state-action space. However, they consider reward maximization and deviate from the practical applications (e.g., these frameworks admit a deterministic optimal policy). Recently, Xiong et al. [72] first formulated the RLHF as a reverse-KL regularized contextual bandit and provided finite-sample guarantees in offline, online, and hybrid settings. We remark that all these papers consider only the reward-based RLHF framework, thus differing from ours.

**Theoretical Study of RLHF under General Preference Oracle.** Our work is related to Dudík et al. [22] and Wang et al. [65]. They investigate preference-based RLHF under a general preference model. The major difference is that we consider the reverse-KL regularized preference, aligning closely with recent LLM advancements [90, 50, 4, 53], while previous work only considers the non-regularized one. Meanwhile, Dudík et al. [22] considers the problem of finite action, while our work and Wang et al. [65] consider the problem with large or even infinite state-action under function approximation. In terms of learning paradigm and algorithmic design, we consider both offline learning from a pre-collected dataset and *batch* online learning with a sparse policy update, while Dudík et al. [22], Wang et al. [65] studies *sequential* online learning that updates policy in each step, which is not feasible in the context of LLMs. Moreover, we demonstrate that the proposed algorithms can be reasonably implemented in practice, but Dudík et al. [22], Wang et al. [65] only focus on information-theoretical algorithms. To summarize, the framework in this work accurately reflects real-world alignment practices thus aligning more closely with the RLHF practice. Our work is closely related to the IPO [3] and Nash learning [46], which also motivate new algorithmic design with a general preference oracle. We comment on the similarities and differences between our framework and theirs as follows. In terms of the problem setting, our work and Nash learning consider the minimax game under the reverse-KL regularized preference, while IPO can be interpreted to find the best response of the fixed reference policy, and may be considered as a special case of the game formulation. In terms of learning paradigm, both the IPO and Nash learning only consider learning toward a *fixed and known* preference oracle, and study the **optimization property** of the problem: how to compute the optimal policy under the given preference oracle. In contrast, we study the **statistical property**, where the preference model needed to be learned and our goal is to find the optimal policy under the underlying ground-truth preference model. In particular, the computational challenge is hidden in Definition 2 and Munos et al. [46] provides a reasonable approximation of the planning oracle. In this sense, our work and Munos et al. [46] are complementary to each other. Finally, the concurrent work Swamy et al. [58] studies the non-regularized general preference model in the *sequential* online setting and aims to find the Nash equilibrium in the context of continuous control tasks. In terms of the observation model, they assume access to the preference score $\mathbb{P}(a^1 \succ a^2 | x, a^1, a^2)$, while we only observe the preference signal $y \sim \mathrm{Ber}(\mathbb{P}(a^1 \succ a^2 | x, a^1, a^2))$. Moreover, they design online RLHF algorithms based on a reduction to the no-regret algorithm like Hedge [25], whose techniques are fundamentally different from ours.

# 8 Conclusion

In this paper, we study the RLHF under a general preference oracle that can capture the non-transitive preferences. Specifically, we formulate the problem as a KL-regularized minimax game between two LLMs, and propose statistically efficient algorithms in both the offline and online settings. The proposed algorithms, with a carefully crafted non-symmetric algorithmic structure, can be practically implemented with reasonable approximations of the information-theoretical computational oracles. We hope our findings can advance the understanding of preference signal modeling in RLHF and stimulate further research beyond the classic reward-based framework.

# 9 Acknowledgment

The authors would like to thank Tianqi Liu for insightful discussions on the training of the preference model, and thank Haoxiang Wang, and Zihao Li for valuable discussions on the preference dataset selection. We also thank Nevena Lazic and Csaba Szepesvari for pointing out a technical gap in the first version.

Wei Xiong and Tong Zhang are partially supported by an NSF IIS grant No. 2416897 and Tong Zhang is partially supported by an NSF IIS grant No. 2416897. Nan Jiang acknowledges funding support from NSF IIS-2112471, NSF CAREER IIS-2141781, Google Scholar Award, and Sloan Fellowship.

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

## A  Authorship and Credit Attribution

All authors provided valuable contributions to this project, each bringing unique expertise and insights that were crucial for its success.

- **CY** investigated the general preference problem, proved the theoretical results for both offline and online settings, and wrote the main part of the paper.
- **WX** first proved the effectiveness of the general preference model, proposed the online iterative algorithm, contributed to the proof and paper writing, and contributed to the experiment.
- **YZ** proved the properties of the general preference problem, made important contributions to the offline result and the proof, contributed to the paper writing.
- **HD** designed the practical implementation under generalized preference model and conducted most experiments to show the effectiveness of the proposed algorithm.
- **NJ** and **TZ** supported and advised the junior authors' works, provided computational resources and suggested experiments and writings.

## B  Notation Table, Related Work, Experimental Details

Table 4: The table of notations used in this paper.

| Notation | Description |
|---|---|
| $\langle z_1, z_2 \rangle$ | The inner product of two vectors $z_1^\top z_2$. |
| $\|z\|_\Sigma$ | The induced norm $\sqrt{z^\top \Sigma z}$. |
| $\mathcal{X}, \mathcal{A}$ | The state (prompt) space and the action (response) space. |
| $\mathbb{P}, P^*$ | The preference oracle defined in Definition 1 and $P^*(x, a^1, a^2) = \mathbb{P}(a^1 \succ a^2 \mid x, a^1, a^2)$. |
| $\mathcal{P}$ | The candidate set of preference model to approximate $P^*$. |
| $y \in \{0, 1\}$ | Preference signal. |
| $\pi, \Pi$ | Policy and policy class. |
| $J(\pi)$ | The KL-regularized target defined in (3). |
| $\eta$ | The coefficient of KL penalty, defined in (3). |
| $\ell_\mathcal{D}$ | The log-likelihood function defined in (7). |
| $d_0$ | Distribution of state (prompt). |
| $\sigma(\cdot)$ | $\sigma(z) = 1/(1 + \exp(-z))$ is the sigmoid function. |
| $\mathcal{C}(\pi, \pi_D, \mathcal{P})$ | Coverage term for version space-based Algorithm 1 defined in Theorem 1. |
| $\widetilde{\mathcal{C}}(\pi, \pi_D, \mathcal{P})$ | Coverage term for uncertainty bonus based Algorithm 3 defined in Theorem 3. |
| $\mathscr{C}((\pi^1, \pi^2), \pi_D, \mathcal{P})$ | Refined coverage term defined in Theorem 4. |
| $\Gamma(\lambda, \pi^1, \pi^2)$ | Information ratio defined in Definition (3). |
| $\widetilde{\Gamma}_t^m(\lambda, \pi^1, \pi^2), \Gamma(x, \pi^1, \pi^2)$ | Uncertainty bonus defined in (11) and (19). |
| $d(\mathcal{P}, \lambda, T)$ | The eluder coefficient defined in Definition 3. |
| $\bar{O}$ | A variant of $O$ that omits logarithmic terms. |

### B.1  More Related Work

**RLHF.** RLHF was first popularized in the deep RL literature by Christiano et al. [14], which served to direct the attention of the RL community to the preference-based feedback, but may further date back to Bennett et al. [8], Knox and Stone [36] in the context of machine learning. It has attracted significant attention recently, mainly due to its tremendous success in Chat-GPT [49]. The most popular and standard RLHF framework is outlined in Ouyang et al. [50], Touvron et al. [60] and we have described the details in Section 1. In terms of reward optimization, PPO [56] is the most well-known algorithm in LLM alignment literature. However, tuning the PPO algorithm to the best performance requires extensive efforts and the result of Chat-GPT4 [49] has not been widely reproduced so far. This motivates another line of works of algorithms that are based on supervised learning. For instance, Dong et al. [18], Yuan et al. [77], Touvron et al. [60], Gulcehre et al. [30], Ji et al. [33] propose reward ranked finetuning, (also known as rejection sampling finetuning), which essentially learns from the best-of-n policy [47] to maximize the reward. The reward-ranked finetuning algorithm is a stable policy optimization algorithm with minimal hyper-parameter configuration and was applied to the RLHF of LLaMA2 [60]. However, it is also observed that the reward ranked finetuning algorithm leads to considerable forgetting in a wide range of tasks (also referred to as the alignment tax), as the algorithmic design only considers reward optimization [60, 42, 13]. One approach to mitigate this issue is to use the KL-regularized formulation, which is widely adopted

in the deep RL approach (e.g. PPO) [90, 67, 50, 4, 37, 41], and other supervised-learning-based algorithms [53, 63, 43, 3], whose theoretical property is studied in Xiong et al. [72]. Among them, (offline) Direct Preference Optimization (DPO) [53] has emerged as an attractive alternative approach to PPO with notable stability and competitive performance. Xiong et al. [72], Hoang Tran [31], Yuan et al. [76] further extend the offline DPO to the iterative (online) variant, and the resulting models demonstrate impressive performance [31, 19]. However, all these algorithms are designed under the reward-based RLHF framework to maximize the underlying reward function (with appropriate regularization).

## B.2    Details of Experiments

**Bradley-Terry model construction.** We follow the previous works [50, 4] to initialize the reward model using an SFT model but replace the last layer with a linear head to predict a scalar score. The loss function of reward modeling is the negative log-likelihood so that minimizing the loss is equivalent to MLE:

$$L_{\mathrm{RM}}(\theta) = -\mathbb{E}_{x,a^w,a^l \sim \mathcal{D}} \log \sigma \big( r_\theta(x, a^w) - r_\theta(x, a^l) \big),$$

where $a^w$ is the preferred response over $a^l$. We train the model for one epoch and use a batch size of 256, a learning rate of $\mathrm{lr} = 1e\text{-}5$, and a cosine learning rate schedule with a warm-up ratio of 0.03.

**Ground-truth preference model for simulation.** Ideally, the $P^*$ is supposed to be a group of human labelers or closed-source LLMs like Chat-GPT. Unfortunately, due to resource constraints, we cannot afford the cost of using these preference oracles. Instead, we follow Gao et al. [26] to use a strong preference model to serve as the $P^*$ in the simulation. Specifically, we adopt the LLaMA3-8B, and train the preference model on a diverse set of open-source preference datasets including HH-RLHF [4], Stanford Human Preferences Dataset (SHP) [23], Ultra-feedback [16], HelpSteer [66], distilabel-capybara[14], distilabel-orca[15], and UltraInteract[16]. Motivated by the Theorem 1 as well as the practical application [50], we include more than 1 comparison pair when a prompt is with more than 2 responses for better coverage. To be specific,

- for SHP, we only use the samples with score ratio $> 2$, and for each prompt, we take at most 5 comparison pairs;

- for HelpSteer, we use all the possible pairs except for those with the same score where the score is averaged over helpfulness and correctness;

- for UltraFeedback, we use all possible pairs except for those with the same score where the score is averaged over all attributes;

- for UltraInteract, we take a subset of 150K pairs into the mixture.

We have about 700K preference pairs in our training stage. We use the package axolotl[17] to perform supervised fine-tuning, with the detailed hyper-parameters given in Appendix B.2. The resulting preference models are evaluated by the reward bench [39], with the results summarized in Table 1. The preference model based on LLaMA3-8B-it achieves state-of-the-art test accuracy and can serve as a stable preference oracle for the simulation study.

We present the hyper-parameters in Table 5. All experiments are conducted on $8 \times$A100-40G with Deepspeed ZeRO-3.

---

[14]`https://huggingface.co/datasets/argilla/distilabel-capybara-dpo-7k-binarized`
[15]`https://huggingface.co/datasets/argilla/distilabel-intel-orca-dpo-pairs`
[16]`openbmb/UltraInteract_pair`
[17]`https://github.com/OpenAccess-AI-Collective/axolotl`

Table 5: Hyper-parameters for reward modeling and preference model construction.

| Models | Hyper-parameter | Value |
|---|---|---|
| Reward model with Gemma-2B-it | Learning rate | $1 \times 10^{-5}$ |
| | Scheduler | Cosine decay with 0.03 warm-up |
| | Epoch | 1 |
| | Batch size | 256 |
| Preference model with Gemma-2B-it | Learning rate | $1 \times 10^{-5}$ |
| | Scheduler | Cosine decay with 0.03 warm-up |
| | Epoch | 1 |
| | Batch size | 128 |
| | Packing | True |
| | Block size | 3072 |
| Preference model with LLaMA3-8B-it | Learning rate | $1 \times 10^{-5}$ |
| | Scheduler | Cosine decay with 0.03 warm-up |
| | Epoch | 1 |
| | Batch size | 128 |
| | Packing | True |
| | Block size | 3072 |

**Examples from Ultra-feedback.** We provide several examples here:

- Create a list of three mistakes to avoid when designing an AI assistant.
- Pretend you're a next.js expert, write ad copy about a free trial on Vercel.
- Can you describe the role of photography in shaping the art world?

## C  Proofs for the Offline Setting

### C.1  Proof for Theorem 1

**Lemma 1.** *Under Assumption 1, with probability at least $1 - \delta$, we have*

$$\sum_{i=1}^{n} (\hat{P}(x_i, a_i^1, a_i^2) - P^*(x_i, a_i^1, a_i^2))^2 \leq \log(|\mathcal{P}|/\delta).$$

*Proof of Lemma 1.* For any fixed function $P \in \mathcal{P}$, we first upper bound its logarithmic moment generating function:

$$\log \mathbb{E} \exp \left( \sum_{i=1}^{n} \log \frac{P(y_i|x_i, a_i^1, a_i^2)}{P^*(y_i|x_i, a_i^1, a_i^2)} \right)$$

$$= \log \mathbb{E} \exp \left( \sum_{i=1}^{n-1} \log \frac{P(y_i|x_i, a_i^1, a_i^2)}{P^*(y_i|x_i, a_i^1, a_i^2)} \right) + \log 2\mathbb{E}_{y_n|x_n, a_n^1, a_n^2} \sqrt{\frac{P(y_n|x_n, a_n^1, a_n^2)}{P^*(y_n|x_n, a_n^1, a_n^2)}}$$

$$= \log \mathbb{E} \exp \left( \sum_{i=1}^{n-1} \log \frac{P(y_i|x_i, a_i^1, a_i^2)}{P^*(y_i|x_i, a_i^1, a_i^2)} \right) + \log \left( 1 - H\big(P(y_n|x_n, a_n^1, a_n^2)\|P^*(y_n|x_n, a_n^1, a_n^2))^2\big) \right)$$

$$\leq \log \mathbb{E} \exp \left( \sum_{i=1}^{n-1} \log \frac{P(y_i|x_i, a_i^1, a_i^2)}{P^*(y_i|x_i, a_i^1, a_i^2)} \right) - H\big(P(y_n|x_n, a_n^1, a_n^2)\|P^*(y_n|x_n, a_n^1, a_n^2))\big)^2$$

$$\leq \ldots \leq -\sum_{i=1}^{n} H\big(P(y_i|x_i, a_i^1, a_i^2)\|P^*(y_i|x_i, a_i^1, a_i^2))\big)^2. \tag{14}$$

We continue to lower-bound the Hellinger by

$$\sum_{i=1}^{n} \Big( H(P(y_i|x_i, a_i^1, a_i^2) \| P^*(y_i|x_i, a_i^1, a_i^2)) \Big)^2$$

$$\geq \sum_{i=1}^{n} \Big( \mathrm{TV}(P(y_i|x_i, a_i^1, a_i^2) \| P^*(y_i|x_i, a_i^1, a_i^2)) \Big)^2$$

$$= \sum_{i=1}^{n} (P(x_i, a_i^1, a_i^2) - P^*(x_i, a_i^1, a_i^2))^2, \tag{15}$$

where the inequality uses the fact that for any distribution $p, q$, $H(p, q) \geq \mathrm{TV}(p, q)$ according to Theorem B.9 of Zhang [83].

Then, by invoking Lemma 6, we obtain for any $P \in \mathcal{P}$, with probability at least $1 - \delta$,

$$\sum_{i=1}^{n} \log \frac{P(y_i|x_i, a_i^1, a_i^2)}{P^*(y_i|x_i, a_i^1, a_i^2)} \leq \log(|\mathcal{P}|/\delta) + \log \mathbb{E} \exp \left( \sum_{i=1}^{n} \log \frac{P(y_i|x_i, a_i^1, a_i^2)}{P^*(y_i|x_i, a_i^1, a_i^2)} \right)$$

$$\leq - \sum_{i=1}^{n} H\Big( P(y_i|x_i, a_i^1, a_i^2) \| P^*(y_i|x_i, a_i^1, a_i^2) \Big)^2 + \log(|\mathcal{P}|/\delta)$$

$$\leq - \sum_{i=1}^{n} (P(x_i, a_i^1, a_i^2) - P^*(x_i, a_i^1, a_i^2))^2 + \log(|\mathcal{P}|/\delta),$$

where the second inequality uses (14), and the last inequality uses (15). By taking $P$ as $\hat{P}$, since $\hat{P}$ is the MLE, we get

$$\sum_{i=1}^{n} (\hat{P}(x_i, a_i^1, a_i^2) - P^*(x_i, a_i^1, a_i^2))^2 \leq \sum_{i=1}^{n} \log \frac{P^*(y_i|x_i, a_i^1, a_i^2)}{P_{\hat{P}}(y_i|x_i, a_i^1, a_i^2)} + \log(|\mathcal{P}|/\delta)$$

$$\leq \log(|\mathcal{P}|/\delta).$$

$\square$

*Proof of Theorem 1.* Let

$$(\hat{\pi}^1, \widetilde{\pi}^2) = \arg \max_{\pi^1 \in \Pi} \arg \min_{\pi^2 \in \Pi} \min_{P \in \widehat{\mathcal{P}}} \mathbb{E}_{x \sim d_0} \mathbb{E}_{a^1 \sim \pi^1, a^2 \sim \pi^2} \Big[ P(x, a^1, a^2) + \eta^{-1} \log \frac{\pi_0(a^1|x)}{\pi^1(a^1|x)} - \eta^{-1} \log \frac{\pi_0(a^2|x)}{\pi^2(a^2|x)} \Big].$$

and use the notation

$$\underline{J}(\pi^1, \pi^2) = \min_{P \in \widehat{\mathcal{P}}} \mathbb{E}_{x \sim d_0} \mathbb{E}_{a^1 \sim \pi^1, a^2 \sim \pi^2} \Big[ P(x, a^1, a^2) + \eta^{-1} \log \frac{\pi_0(a^1|x)}{\pi^1(a^1|x)} - \eta^{-1} \log \frac{\pi_0(a^2|x)}{\pi^2(a^2|x)} \Big].$$

Let $\widetilde{\pi}^2_* = \min_{\pi^2 \in \Pi} \underline{J}(\pi^1_*, \pi^2)$ and $\pi^{\dagger,2} = \min_{\pi^2 \in \Pi} J(\hat{\pi}^1, \pi^2)$. The following decomposition holds

$$J(\pi^1_*, \pi^2_*) - J(\hat{\pi}^1, \pi^{\dagger,2}) \leq \underbrace{J(\pi^1_*, \pi^2_*) - J(\pi^1_*, \widetilde{\pi}^2_*)}_{q_1} + \underbrace{J(\pi^1_*, \widetilde{\pi}^2_*) - \underline{J}(\pi^1_*, \widetilde{\pi}^2_*)}_{q_2} + \underbrace{\underline{J}(\pi^1_*, \widetilde{\pi}^2_*) - \underline{J}(\hat{\pi}^1, \widetilde{\pi}^2)}_{q_3}$$

$$+ \underbrace{\underline{J}(\hat{\pi}^1, \widetilde{\pi}^2) - \underline{J}(\hat{\pi}^1, \pi^{\dagger,2})}_{q_4} + \underbrace{\underline{J}(\hat{\pi}^1, \pi^{\dagger,2}) - J(\hat{\pi}^1, \pi^{\dagger,2})}_{q_5}.$$

Then, we bound these terms separately. For the term $q_1$, since $(\pi^1_*, \pi^2_*)$ is the Nash equilibrium of $J$, we have $q_1 \leq 0$. For the term $q_2$,

$$q_2 = \mathbb{E}_{x \sim d_0} \mathbb{E}_{a^1 \sim \pi^1_*, a^2 \sim \widetilde{\pi}^2_*} P^*(x, a^1, a^2) - \min_{P \in \widehat{\mathcal{P}}} \mathbb{E}_{x \sim d_0} \mathbb{E}_{a^1 \sim \pi^1_*, a^2 \sim \widetilde{\pi}^2_*} P(x, a^1, a^2)$$

$$= \min_{P \in \widehat{\mathcal{P}}} \mathbb{E}_{x \sim d_0} \mathbb{E}_{a^1 \sim \pi^1_*, a^2 \sim \widetilde{\pi}^2_*} [\hat{P}(x, a^1, a^2) - P(x, a^1, a^2)] + \mathbb{E}_{x \sim d_0} \mathbb{E}_{a^1 \sim \pi^1_*, a^2 \sim \hat{\pi}^2_*} [P^*(x, a^1, a^2) - \hat{P}(x, a^1, a^2)]$$

$$\leq 2\beta \widetilde{\Gamma}(\pi^1_*, \widetilde{\pi}^2_*),$$

where we define

$$\widetilde{\Gamma}(\pi^1, \pi^2) := \sup_{P \in \widehat{\mathcal{P}}} \frac{|\mathbb{E}_{x \sim d_0}[P(x, \pi^1, \pi^2) - \hat{P}(x, \pi^1, \pi^2)]|}{\sqrt{\lambda + \sum_{i=1}^n (P(x_i, a_i^1, a_i^2) - \hat{P}(x_i, a_i^1, a_i^2))^2}}.$$

By the optimality of $\hat{\pi}^1$, term $q_3 \leq 0$. Since $\widetilde{\pi}^2$ is the best response to $\hat{\pi}^1$ with respect to $\underline{J}$, we have $q_4 \leq 0$. From Lemma 1, we know that $P^\star \in \widehat{\mathcal{P}}$, thus $q_5 \leq 0$. Putting everything together, we obtain that

$$J(\pi_*^1, \pi_*^2) - J(\hat{\pi}^1, \pi^{\dagger,2}) \leq 2\beta\widetilde{\Gamma}(\pi_*^1, \widetilde{\pi}_*^2). \tag{16}$$

Then, by invoking Lemma 8 with a union bound over $P \in \mathcal{P}$, with probability at least $1 - \delta$, we obtain that

$$0.5n\mathbb{E}_{x \sim d_0, a^1 \sim \pi_D^1, a^2 \sim \pi_D^2}(P(x, a^1, a^2) - \hat{P}(x, a^1, a^2))^2$$

$$\leq \sum_{i=1}^n (P(x_i, a_i^1, a_i^2) - \hat{P}(x_i, a_i^1, a_i^2))^2 + \log(|\mathcal{P}|/\delta),$$

which implies that with probability at least $1 - \delta$,

$$\widetilde{\Gamma}(\pi_*^1, \widetilde{\pi}_*^2)$$

$$= \sup_{P \in \mathcal{P}} \frac{|\mathbb{E}_{x \sim d_0}[P(x, \pi_*^1, \widetilde{\pi}_*^2) - \hat{P}(x, \pi_*^1, \widetilde{\pi}^2)]|}{\sqrt{\lambda + \sum_{i=1}^n (P(x_i, a_i^1, a_i^2) - \hat{P}(x_i, a_i^1, a_i^2))^2}}$$

$$\leq \sup_{P \in \mathcal{P}} \frac{|\mathbb{E}_{x \sim d_0}[P(x, \pi_*^1, \widetilde{\pi}_*^2) - \hat{P}(x, \pi_*^1, \widetilde{\pi}_*^2)]|}{\sqrt{\lambda - \log(|\mathcal{P}|/\delta) + 0.5n\mathbb{E}_{x \sim d_0, a^1 \sim \pi_D^1, a^2 \sim \pi_D^2}(P(x, a^1, a^2) - \hat{P}(x, a^1, a^2))^2}}$$

$$= \sqrt{\frac{2}{n}} \sup_{P \in \mathcal{P}} \frac{|\mathbb{E}_{x \sim d_0}[P(x, \pi_*^1, \widetilde{\pi}_*^2) - \hat{P}(x, \pi_*^1, \widetilde{\pi}_*^2)]|}{\sqrt{\mathbb{E}_{x \sim d_0, a^1 \sim \pi_D^1, a^2 \sim \pi_D^2}(P(x, a^1, a^2) - \hat{P}(x, a^1, a^2))^2}}$$

$$= \sqrt{\frac{2\mathcal{C}(\pi_*^1, \pi_D, \mathcal{P})}{n}}.$$

Hence, we complete the proof. $\qquad\square$

## C.2 Learning with Pessimism via Uncertainty Bonus

In this subsection, we introduce another offline algorithm, Pessimistic Equilibrium Learning from Human Feedback (PELHF) with Uncertainty Bonus in Algorithm 3. Given an offline dataset $\mathcal{D}_{\text{off}}$, we first obtain the maximum likelihood estimation (MLE) by maximizing (7). Then, we take the lower confidence bound (LCB) for the max-player as the preference estimations by subtracting a bonus function $\beta\Gamma(\cdot, \cdot, \cdot)$:

$$\underline{J}(x, \pi^1, \pi^2) = \mathbb{E}_{a^1 \sim \pi^1, a^2 \sim \pi^2}\left[\hat{P}(x, a^1, a^2) - \beta\Gamma(x, a^1, a^2) + \eta^{-1}\log\frac{\pi_0(a^1|x)}{\pi^1(a^1|x)} - \eta^{-1}\log\frac{\pi_0(a^2|x)}{\pi^2(a^2|x)}\right]. \tag{17}$$

Then, we obtain the policy $\hat{\pi}^1$ by solving the minimax problems with $\underline{J}$. We now discuss how to construct the bonus function to ensure pessimism.

**Bonus Construction.** The bonus function $\Gamma : \mathcal{X} \times \mathcal{A} \times \mathcal{A} \to \mathbb{R}^+$ serves to control the *point-wise* confidence interval so that with high probability, $\hat{P}(x, a^1, a^2) - \beta\Gamma(x, a^1, a^2) \leq P^*(x, a^1, a^2) \leq \hat{P}(x, a^1, a^2) + \beta\Gamma(x, a^1, a^2)$ holds for any $(x, a^1, a^2)$. To this end, we construct the bonus as the ratio between the *out-of-sample* error and the *in-sample* error on the preference dataset $\mathcal{D}_{\text{off}}$:

$$\Gamma(x, \pi^1, \pi^2) = \sup_{P \in \mathcal{P}} \frac{|P(x, \pi^1, \pi^2) - \hat{P}(x, \pi^1, \pi^2)|}{\sqrt{\lambda + \sum_{i=1}^n (P(x_i, a_i^1, a_i^2) - \hat{P}(x_i, a_i^1, a_i^2))^2}}, \tag{19}$$

where we also set $\beta$ as an upper bound of the $\lambda$-regularized in-sample error. This uncertainty is also characterized by the relative preference function class and shares a similar spirit with the information ratio considered in Zhang [83], Ye et al. [74, 75], which depicts the uncertainty with respect to the value function class. Also see Definition 3 for a more detailed illustration.

---

**Algorithm 3** Pessimistic Equilibrium Learning from Human Feedback with Uncertainty Bonus

---

1: **Input:** Dataset $\mathcal{D}_{\text{off}} = \{(x_i, a_i^1, a_i^2, y_i)\}_{i=1}^n$, preference space $\mathcal{P}$, policy class $\Pi$, parameter $\eta, \beta > 0$.
2: Compute the MLE $\hat{P}$ with $\ell_{\mathcal{D}_{\text{off}}}$ defined in (7) and construct bonus as in (19).
3: Compute the best policy under conservative estimation

$$\hat{\pi}^1(\cdot|x) = \underset{\pi^1 \in \Pi}{\operatorname{argmax}} \min_{\pi^2 \in \Pi} \mathbb{E}_{a^1 \sim \pi^1, a^2 \sim \pi^2}\left[\hat{P}(x, a^1, a^2) - \beta\Gamma(x, \pi^1, \pi^2) + \eta^{-1}\log\frac{\pi_0(a^1|x)}{\pi^1(a^1|x)} - \eta^{-1}\log\frac{\pi_0(a^2|x)}{\pi^2(a^2|x)}\right].$$

(18)

4: **Output:** $\hat{\pi}^1$.

---

### C.2.1 Analysis for Algorithm 3

Now, we are ready to present the suboptimality bound of $\hat{\pi}^1$ from Algorithm 3.

**Theorem 3.** *If we set $\lambda = \log(|\mathcal{P}|/\delta)$ and $\beta^2 = 2\log(|\mathcal{P}|/\delta)$, then, with probability at least $1 - \delta$, the output policy of Algorithm 3 satisfies*

$$J(\pi_1^*, \pi_2^*) - J(\hat{\pi}^1, \dagger) \leq 4\beta\sqrt{\frac{\widetilde{\mathcal{C}}(\pi_*^1, \pi_D, \mathcal{P})}{n}} - \mathbb{E}_{x \sim d_0}\left[\eta^{-1}D_{\text{KL}}(\pi_*^1(\cdot|x)\|\hat{\pi}^1(\cdot|x))\right].$$

*where*

$$\widetilde{\mathcal{C}}(\pi_*^1, \pi_D, \mathcal{P}) = \max_{\pi^2 \in \Pi} \mathbb{E}_{x \sim d_0} \sup_{\hat{P} \in \mathcal{P}} \frac{(P(x, \pi_*^1, \pi^2) - \hat{P}(x, \pi_*^1, \pi^2))^2}{\mathbb{E}_{x \sim d_0, a^1 \sim \pi_D^1, a^2 \sim \pi_D^2}(P(x, a^1, a^2) - \hat{P}(x, a^1, a^2))^2}.$$

*Proof.* Recall that our pessimistic value estimations are

$$\underline{J}(x, \pi^1, \pi^2) = \mathbb{E}_{a^1 \sim \pi^1, a^2 \sim \pi^2}\left[\hat{P}(x, a^1, a^2) - \beta\Gamma(x, a^1, a^2) + \eta^{-1}\log\frac{\pi_0(a^1|x)}{\pi^1(a^1|x)} - \eta^{-1}\log\frac{\pi_0(a^2|x)}{\pi^2(a^2|x)}\right].$$

For convenience, we also use the notation

$$J(x, \pi^1, \pi^2) = \mathbb{E}_{a^1 \sim \pi^1, a^2 \sim \pi^2}\left[P^*(x, a^1, a^2) + \eta^{-1}\log\frac{\pi_0(a^1|x)}{\pi^1(a^1|x)} - \eta^{-1}\log\frac{\pi_0(a^2|x)}{\pi^2(a^2|x)}\right].$$

We decompose the suboptimality gap of $\hat{\pi}^1$ at prompt $x$ as follows:

$$J(x, \pi_*^1, \pi_*^2) - J(x, \hat{\pi}^1, \dagger) \leq \underbrace{\underline{J}(x, \hat{\pi}^1, \widetilde{\pi}^2) - J(x, \hat{\pi}^1, \dagger)}_{p_1} + \underbrace{\underline{J}(x, \pi_*^1, \widetilde{\pi}^2) - \underline{J}(x, \hat{\pi}^1, \widetilde{\pi}^2)}_{p_2}$$

$$+ \underbrace{J(x, \pi_*^1, \widetilde{\pi}^2) - \underline{J}(x, \pi_*^1, \widetilde{\pi}^2)}_{p_3} + \underbrace{J(x, \pi_*^1, \pi_*^2) - J(x, \pi_*^1, \widetilde{\pi}^2)}_{p_4}.$$

We proceed based on assuming the following event holds:

$$\sum_{i=1}^n (\hat{P}(x_i, a_i^1, a_i^2) - P^*(x_i, a_i^1, a_i^2))^2 \leq \beta^2/2.$$

For the term $p_1$, we have

$$p_1 = \underline{J}(x, \hat{\pi}^1, \widetilde{\pi}^2) - \min_{\pi^2}\left\{P^*(x, \hat{\pi}^1, \pi^2) - \eta^{-1}D_{\text{KL}}(\hat{\pi}^1(\cdot|x)\|\pi_0(\cdot|x)) + \eta^{-1}D_{\text{KL}}(\pi^2(\cdot|x)\|\pi_0(\cdot|x))\right\}$$

$$= \underline{J}(x, \hat{\pi}^1, \widetilde{\pi}^2) - \min_{\pi^2}\left\{P^*(x, \hat{\pi}^1, \pi^2) - \hat{P}(x, \hat{\pi}^1, \pi^2) + \hat{P}(x, \hat{\pi}^1, \pi^2) - \eta^{-1}D_{\text{KL}}(\hat{\pi}^1(\cdot|x)\|\pi_0(\cdot|x)) + \eta^{-1}D_{\text{KL}}(\pi^2(\cdot|x)\|\pi_0$$

$$\leq \underline{J}(x, \hat{\pi}^1, \widetilde{\pi}^2) - \min_{\pi^2}\left\{\hat{P}(x, \hat{\pi}^1, \pi^2) - \beta\Gamma(x, \hat{\pi}^1, \pi^2) - \eta^{-1}D_{\text{KL}}(\hat{\pi}^1(\cdot|x)\|\pi_0(\cdot|x)) + \eta^{-1}D_{\text{KL}}(\pi^2(\cdot|x)\|\pi_0(\cdot|x))\right\}$$

$$= 0,$$

where the inequality is because

$$P^*(x, \hat{\pi}^1, \pi^2) - \hat{P}(x, \hat{\pi}^1, \pi^2)$$

$$\geq -\sqrt{\lambda + \sum_{i=1}^n (P^*(x_i, a_i^1, a_i^2) - \hat{P}(x_i, a_i^1, a_i^2))^2} \cdot \sup_{P' \in \mathcal{P}} \frac{|\mathbb{E}_{a^1 \sim \hat{\pi}^1, a_2 \sim \pi^2}[P'(x, a^1, a^2) - \hat{P}(x, a^1, a^2)]|}{\sqrt{\lambda + \sum_{i=1}^n (P'(x_i, a_i^1, a_i^2) - \hat{P}(x_i, a_i^1, a_i^2))^2}}$$

$$\geq -\beta \Gamma(x, \hat{\pi}^1, \pi^2).$$

Here the last step uses Lemma 1 to bound the in-sample error. For the term $p_2$, we can write it as

$$p_2 = \hat{P}(x, \pi_*^1, \widetilde{\pi}^2) - \beta \Gamma(x, \pi_*^1, \widetilde{\pi}^2) - \hat{P}(x, \hat{\pi}^1, \widetilde{\pi}^2) + \beta \Gamma(x, \hat{\pi}^1, \widetilde{\pi}^2)$$
$$- \eta^{-1} D_{\mathrm{KL}}(\pi_*^1(\cdot|x) \| \pi_0(\cdot|x)) + \eta^{-1} D_{\mathrm{KL}}(\hat{\pi}^1(\cdot|x) \| \pi_0(\cdot|x)).$$

We note that

$$\hat{\pi}^1 = \operatorname*{argmax}_{\pi^1} \underline{J}(x, \pi^1, \widetilde{\pi}^2)$$

$$= \operatorname*{argmax}_{\pi^1} \mathbb{E}_{a^1 \sim \pi^1, a^2 \sim \widetilde{\pi}^2} \left[ \hat{P}(x, a^1, a^2) - \beta \Gamma(x, a^1, a^2) + \eta^{-1} \log \frac{\pi_0(a^1|x)}{\pi^1(a^1|x)} \right].$$

Therefore, we can invoke Lemma 9 with $\pi = \pi_*^1$, $\hat{\pi} = \hat{\pi}^1$, and $\hat{P}(x, a) = \hat{P}(x, a, \widetilde{\pi}^2) - \beta \Gamma(x, a, \widetilde{\pi}^2)$ to obtain

$$\hat{P}(x, \pi_*^1, \widetilde{\pi}^2) - \beta \Gamma(x, \pi_*^1, \widetilde{\pi}^2) - \hat{P}(x, \hat{\pi}^1, \widetilde{\pi}^2) + \beta \Gamma(x, \hat{\pi}^1, \widetilde{\pi}^2)$$
$$+ \eta^{-1} D_{\mathrm{KL}}(\hat{\pi}^1(\cdot|x) \| \pi_0(\cdot|x)) - \eta^{-1} D_{\mathrm{KL}}(\pi_*^1(\cdot|x) \| \pi_0(\cdot|x))$$
$$= -\eta^{-1} D_{\mathrm{KL}}(\pi_*^1(\cdot|x) \| \hat{\pi}^1(\cdot|x)),$$

which implies that

$$p_2 = -\eta^{-1} D_{\mathrm{KL}}(\pi_*^1(\cdot|x) \| \hat{\pi}^1(\cdot|x)).$$

For the term $p_3$, we can also get from Lemma 1 that

$$p_3 = P^*(x, \pi_*^1, \widetilde{\pi}^2) - \hat{P}(x, \pi_*^1, \widetilde{\pi}^2) + \beta \Gamma(x, \pi_*^1, \widetilde{\pi}^2)$$
$$\leq 2\beta \Gamma(x, \pi_*^1, \widetilde{\pi}^2).$$

According to Lemma 5, since $\pi_*^2$ is the best response to $\pi_*^1$ with respect to $J(x, \cdot, \cdot)$, we have $p_4 \leq 0$. Putting everything together, we have with probability at least $1 - \delta$,

$$J(x, \pi_*^1, \pi_*^2) - J(x, \hat{\pi}^1, \dagger) \leq 2\beta \Gamma(x, \pi_*^1, \widetilde{\pi}^2) - \eta^{-1} D_{\mathrm{KL}}(\pi_*^1(\cdot|x) \| \hat{\pi}^1(\cdot|x)).$$

Similar to the proof of Theorem 1, we invoke Lemma 8 with a union bound over $P \in \mathcal{P}$ and obtain that with probability at least $1 - \delta$,

$$0.5 n \mathbb{E}_{x \sim d_0, a^1 \sim \pi_D^1, a^2 \sim \pi_D^2}(P(x, a^1, a^2) - \hat{P}(x_s, a_s^1, a_s^2))^2$$

$$\leq \sum_{i=1}^n (P(x_i, a_i^1, a_i^2) - P^*(x_i, a_i^1, a_i^2))^2 + \log(|\mathcal{P}|/\delta),$$

which implies that probability at least $1 - \delta$,

$$\mathbb{E}_{x \sim d_0} \Gamma(x, \pi_*^1, \widetilde{\pi}^2)$$

$$= \mathbb{E}_{x \sim d_0} \sup_{P \in \mathcal{P}} \frac{|P(x, \pi_*^1, \widetilde{\pi}^2) - \hat{P}(x, \pi_*^1, \widetilde{\pi}^2)|}{\sqrt{\lambda + \sum_{i=1}^n (P(x_i, a_i^1, a_i^2) - \hat{P}(x_i, a_i^1, a_i^2))^2}}$$

$$\leq \mathbb{E}_{x \sim d_0} \sup_{P \in \mathcal{P}} \frac{|P(x, \pi_*^1, \widetilde{\pi}^2) - \hat{P}(x, \pi_*^1, \widetilde{\pi}^2)|}{\sqrt{\lambda - \log(|\mathcal{P}|/\delta) + 0.5 n \mathbb{E}_{x \sim d_0, a^1 \sim \pi_D^1, a^2 \sim \pi_D^2}(P(x, a^1, a^2) - \hat{P}(x_s, a^1, a))^2}}$$

$$= \sqrt{\frac{2}{n}} \mathbb{E}_{x \sim d_0} \sup_{P \in \mathcal{P}} \frac{|P(x, \pi_*^1, \widetilde{\pi}^2) - \hat{P}(x, \pi_*^1, \widetilde{\pi}^2)|}{\sqrt{\mathbb{E}_{x \sim d_0, a^1 \sim \pi_D^1, a^2 \sim \pi_D^2}(P(x_s, a^1, a^2) - \hat{P}(x, a^1, a^2))^2}}$$

$$\leq \sqrt{\frac{2\widetilde{\mathcal{C}}(\pi_*^1, \pi_D, \mathcal{P})}{n}},$$

where the second equality holds due to $\lambda = \log(|\mathcal{P}|/\delta)$. Therefore, we complete the proof. $\qquad\square$

**Comparison between Bonus and Version Space.** Compared to the bound in Theorem 1, the bound in Theorem 3 enjoys an additional negative KL divergence term between $\pi_*^1$ and $\hat{\pi}^1$. Both Theorem 1 and Theorem 3 depend on a distribution-shift term between Nash policy $\pi_*^1$ and the policy $\pi_D$ that the data is complied with. The difference is that Theorem 1 enjoys a sharper term $\mathcal{C}$ because of Jensen's inequality and the expectations are inside the sup operator. In terms of applicability, the version-space-based pessimism is preferred because it does not require a point-wise uncertainty estimator, thus applying to general cases. In contrast, point-wise pessimism, or more generally, optimism/pessimism via a biased target may be easier to heuristically approximate in practice, as shown in Coste et al. [15], Xie et al. [69], Zhang [82], Liu et al. [44].

## C.3 Analysis for Refined Coverage Condition

In this subsection, we show that with an improved analysis, Algorithm 1 enjoys a refined coverage condition, similar to the coverage notion in [84].

**Theorem 4.** *If Assumption 1 holds, and we set $\lambda = \log(|\mathcal{P}|/\delta)$ and $\beta^2 = 2\log(|\mathcal{P}|/\delta)$, then, with probability at least $1 - \delta$, the output policy of Algorithm 1 satisfies*

$$J(\pi_1^*, \pi_2^*) - J(\hat{\pi}^1, \dagger) \leq \min_{\pi^2 \in \Pi} 4\beta\sqrt{\frac{\mathscr{C}((\pi_*^1, \pi^2), \pi_D, \mathcal{P})}{n}} + \text{subopt}^{\pi_*^1, \widetilde{\pi}_*^2}(\pi^2),$$

*where*

$$\mathscr{C}((\pi_*^1, \pi^2), \pi_D, \mathcal{P}) = \sup_{P \in \mathcal{P}} \frac{(\mathbb{E}_{x \sim d_0, a^1 \sim \pi_*^1, a^2 \sim \pi^2}[P(x, a^1, a^2) - \hat{P}(x, a^1, a^2)])^2}{\mathbb{E}_{x \sim d_0, a^1 \sim \pi_D^1, a^2 \sim \pi_D^2}(P(x, a^1, a^2) - \hat{P}(x, a^1, a^2))^2},$$

$$\text{subopt}^{\pi_*^1, \widetilde{\pi}_*^2}(\pi^2) = \underline{J}(\pi_*^1, \pi^2) - \underline{J}(\pi_*^1, \widetilde{\pi}_*^2).$$

*Proof.* Recall that

$$\underline{J}(\pi^1, \pi^2) = \min_{P \in \widehat{\mathcal{P}}} \mathbb{E}_{x \sim d_0} \mathbb{E}_{a^1 \sim \pi^1, a^2 \sim \pi^2}\left[P(x, a^1, a^2) + \eta^{-1} \log \frac{\pi_0(a^1|x)}{\pi^1(a^1|x)} - \eta^{-1} \log \frac{\pi_0(a^2|x)}{\pi^2(a^2|x)}\right].$$

and $\widetilde{\pi}_*^2 = \min_{\pi^2 \in \Pi} \underline{J}(\pi_*^1, \pi^2)$ and $\pi^{\dagger,2} = \min_{\pi^2 \in \Pi} J(\hat{\pi}^1, \pi^2)$. We observe that for any $\pi^2$, the following decomposition holds

$$J(\pi_*^1, \pi_*^2) - J(\hat{\pi}^1, \pi^{\dagger,2}) \leq \underbrace{J(\pi_*^1, \pi_*^2) - J(\pi_*^1, \pi^2)}_{q_1} + \underbrace{J(\pi_*^1, \pi^2) - \underline{J}(\pi_*^1, \pi^2)}_{q_2} + \underbrace{\underline{J}(\pi_*^1, \pi^2) - \underline{J}(\pi_*^1, \widetilde{\pi}_*^2)}_{q_3}$$

$$+ \underbrace{\underline{J}(\pi_*^1, \widetilde{\pi}_*^2) - \underline{J}(\hat{\pi}^1, \widetilde{\pi}^2)}_{q_4} + \underbrace{\underline{J}(\hat{\pi}^1, \widetilde{\pi}^2) - \underline{J}(\hat{\pi}^1, \pi^{\dagger,2})}_{q_5} + \underbrace{\underline{J}(\hat{\pi}^1, \pi^{\dagger,2}) - J(\hat{\pi}^1, \pi^{\dagger,2})}_{q_6}.$$

For the term $q_1$, since $(\pi_*^1, \pi_*^2)$ is the Nash equilibrium of $J$, we have $q_1 \leq 0$. By the optimality of $\hat{\pi}^1$, term $q_4 \leq 0$. From the proof of Theorem 1, we know that $q_5 \leq 0$ and $q_6 \leq 0$. The term $q_3 = \underline{J}(\pi_*^1, \pi^2) - \underline{J}(\pi_*^1, \widetilde{\pi}_*^2) := \text{subopt}^{\pi_*^1, \widetilde{\pi}_*^2}(\pi^2)$ measures the suboptimality gap between $\pi^2$ and $\widetilde{\pi}_*^2$ under the pessimistic estimation $\underline{J}$ and Nash policy $\pi_*^1$. For the term $q_2$, we have

$$q_4 = \mathbb{E}_{x \sim d_0} \mathbb{E}_{a^1 \sim \pi_*^1, a^2 \sim \pi^2} P^*(x, a^1, a^2) - \min_{P \in \widehat{\mathcal{P}}} \mathbb{E}_{x \sim d_0} \mathbb{E}_{a^1 \sim \pi_*^1, a^2 \sim \pi^2} P(x, a^1, a^2)$$

$$= \min_{P \in \widehat{\mathcal{P}}} \mathbb{E}_{x \sim d_0} \mathbb{E}_{a^1 \sim \pi_*^1, a^2 \sim \pi^2} [\hat{P}(x, a^1, a^2) - P(x, a^1, a^2)] + \mathbb{E}_{x \sim d_0} \mathbb{E}_{a^1 \sim \pi_*^1, a^2 \sim \pi^2} [P^*(x, a^1, a^2) - \hat{P}(x, a^1, a^2)]$$

$$\leq 2\beta\widetilde{\Gamma}(\pi_*^1, \pi^2).$$

Therefore, we obtain that

$$J(\pi_*^1, \pi_*^2) - J(\hat{\pi}^1, \pi^{\dagger,2}) \leq 2\beta\widetilde{\Gamma}(\pi_*^1, \pi^2) + \text{subopt}^{\pi_*^1, \widetilde{\pi}_*^2}(\pi^2). \tag{20}$$

Since Equation (20) holds for any $\pi_2$, we further have

$$J(\pi_*^1, \pi_*^2) - J(\hat{\pi}^1, \pi^{\dagger,2}) \leq \min_{\pi^2 \in \Pi} 2\beta\widetilde{\Gamma}(\pi_*^1, \pi^2) + \text{subopt}^{\pi_*^1, \widetilde{\pi}_*^2}(\pi^2).$$

The proof for bounding $\widetilde{\Gamma}(\pi_*^1, \pi^2)$ is the same as that of Theorem 1. $\qquad\square$

We can prove that Algorithm 3 also enjoys a similar bound and coverage condition. We now provide a breakdown of the terms in Theorem 4.

- First, we can simply let $\pi^2 = \widetilde{\pi}_*^2$, the best response to $\pi_*^1$ under the pessimistic estimation, and then the bound becomes $4\beta\sqrt{\mathscr{C}((\pi_*^1, \widetilde{\pi}_*^2), \pi_D, \mathcal{P})/n}$, which measures the coverage of the dataset $\mathcal{D}$ on $(\pi_*^1, \widetilde{\pi}_*^2)$. When the distribution of $\mathcal{D}$ aligns well with the distribution induced by $(\pi_*^1, \widetilde{\pi}_*^2)$, the dataset has a good coverage on $(\pi_*^1, \widetilde{\pi}_*^2)$ and the term $\mathscr{C}((\pi_*^1, \widetilde{\pi}_*^2), \pi_D, \mathcal{P})$ becomes small.
- When $\mathcal{D}$ has a poor coverage on $(\pi_*^1, \widetilde{\pi}_*^2)$, i.e., $\mathscr{C}((\pi_*^1, \widetilde{\pi}_*^2), \pi_D, \mathcal{P})$ is large, our bound adapts to an alternate policy $\pi^2$ that achieves a better trade-off between the suboptimality term $\mathrm{subopt}^{\pi_*^1, \widetilde{\pi}_*^2}(\pi^2)$ and the coverage term $\mathscr{C}((\pi_*^1, \pi^2), \pi_D, \mathcal{P})$. Here the suboptimality term measures the performance gap between $\pi^2$ and $\widetilde{\pi}_*^2$ under $\underline{J}(\pi_*^1, \cdot)$.

**Comparison to Unilateral Coverage.** The unilateral coverage [17, 86] requires the dataset to cover all unilateral pairs $(\pi_*^1, \pi^2)$ for any $\pi^2 \in \Pi$, making the bound in Theorem 1 depend on the coverage term of the worst pair. In contrast, the bound in Theorem 4 automatically adapts to the best $\pi^2$, achieving the trade-off between the coverage term and the suboptimality term, thus offering a more flexible coverage condition.

# D   Proofs for the Online Setting

*Proof of Theorem 2.* We start with the in-sample error estimation. Similar to the proof of Lemma 1 but with an additional union bound over $t \in [T]$, we have with probability at least $1 - \delta/2$, for any $t \in [T]$,

$$\frac{1}{m}\sum_{i=1}^{m}(\hat{P}_t(x_{t,i}, a_{t,i}^1, a_{t,i}^2) - P^*(x_{t,i}, a_{t,i}^1, a_{t,i}^2))^2 \le \frac{\log(2T|\mathcal{P}|/\delta)}{m},$$

which implies that we can set $\beta^2 = \frac{T\log(2T|\mathcal{P}|/\delta)}{m}$ so that $\beta\widetilde{\Gamma}_t^m$ is a valid uncertainty bonus:

$$\mathbb{E}_x\hat{P}_t(x, \pi^1, \pi^2) - \beta\widetilde{\Gamma}_t^m(\lambda, \pi^1, \pi^2) \le \mathbb{E}_x P^*(x, \pi^1, \pi^2) \le \mathbb{E}_x\hat{P}_t(x, \pi^1, \pi^2) - \beta\widetilde{\Gamma}_t^m(\lambda, \pi^1, \pi^2). \quad (21)$$

We proceed to prove that there exists at least one iteration, the out-of-sample error is close to the in-sample error. According to the Definition 3, we know that for any sequence $\{(\hat{\pi}_t^1, \hat{\pi}_t^2)\}_{t=1}^T$, it holds that

$$\sum_{t=1}^{T} \min\left(1, (\Gamma_t(\lambda, \hat{\pi}_t^1, \hat{\pi}_t^2))^2\right) \le d(\mathcal{P}, \lambda, T).$$

Since each term on the left-hand side is non-negative, we know that there exists at least a $t_0 \in [T]$ such that the value at $t_0$ is smaller or equal to the average value:

$$\min\left(1, (\Gamma_{t_0}(\lambda, \hat{\pi}_{t_0}^1, \hat{\pi}_{t_0}^2))^2\right) \le \frac{d(\mathcal{P}, \lambda, T)}{T} \le \frac{1}{2},$$

which further implies that $(\Gamma_{t_0}(\lambda, \hat{\pi}_{t_0}^1, \hat{\pi}_{t_0}^2))^2 \le \frac{1}{2}$.

We use the notation $\widetilde{\pi}_t^2 = \mathrm{argmin}_{\pi'} J(\hat{\pi}_t^1, \pi')$ and

$$\hat{J}(x, \pi^1, \pi^2) = \hat{P}(x, \pi^1, \pi^2) - \eta^{-1}D_{\mathrm{KL}}(\pi^1(a^1|x)\|\pi_0(a^1|x)) + \eta^{-1}D_{\mathrm{KL}}(\pi^2(a^1|x)\|\pi_0(a^1|x)).$$

For each $t \in [T]$, the suboptimality for the max-player can be written as

$$J(\pi_1^*, \pi_2^*) - J(\hat{\pi}_t^1, \dagger)$$

$$= \mathbb{E}_{x\sim d_0}\left[J(x, \pi^*, \pi^*) - \hat{J}(x, \hat{\pi}_t^1, \widetilde{\pi}_t^2) + \hat{J}(x, \hat{\pi}_t^1, \widetilde{\pi}_t^2) - J(x, \hat{\pi}_t^1, \widetilde{\pi}_t^2)\right]$$

$$\le \mathbb{E}_{x\sim d_0}\left[-\hat{P}(x, \hat{\pi}_t^1, \widetilde{\pi}_t^2) + \eta^{-1}D_{\mathrm{KL}}(\hat{\pi}_t^1(\cdot|x)\|\pi_0(\cdot|x)) - \eta^{-1}D_{\mathrm{KL}}(\widetilde{\pi}_t^2(\cdot|x)\|\pi_0(\cdot|x))\right] + \beta\widetilde{\Gamma}_t^m(\lambda, \hat{\pi}_t^1, \widetilde{\pi}_t^2)$$

$$\le \mathbb{E}_{x\sim d_0}\left[-\hat{P}(x, \hat{\pi}_t^1, \hat{\pi}_t^1) + \eta^{-1}D_{\mathrm{KL}}(\hat{\pi}_t^1(\cdot|x), \pi_0(\cdot|x)) - \eta^{-1}D_{\mathrm{KL}}(\hat{\pi}_t^1(\cdot|x), \pi_0) - \eta^{-1}D_{\mathrm{KL}}(x, \widetilde{\pi}_t^2, \hat{\pi}_t^1)\right]$$

$$\quad + \beta\widetilde{\Gamma}_t^m(\lambda, \hat{\pi}_t^1, \hat{\pi}_t^2)$$

$$= \beta\widetilde{\Gamma}_t^m(\lambda, \hat{\pi}_t^1, \hat{\pi}_t^2) - \eta^{-1}\mathbb{E}_{x\sim d_0}D_{\mathrm{KL}}(\widetilde{\pi}_t^2(\cdot|x)\|\hat{\pi}_t^1(\cdot|x)),$$

where the first inequality uses (21) and $J^*(x) = J^*(x, \pi^*, \pi^*) = 0$, in the second inequality we use the definition of $\hat{\pi}_t^2$, and $\hat{P}(x, \pi, \pi) = 0$ in the last equality.

We proceed to connect the empirical bonus with the information ratio. Combining Lemma 8 with a union bound over $(P, s) \in \mathcal{P} \times [T]$, with probability at least $1 - \delta/2$, we know that

$$0.5\mathbb{E}_{x_s \sim d_0, a_s^1 \sim \hat{\pi}_s^1, a_s^2 \sim \hat{\pi}_s^2}(P(x_s, a_s^1, a_s^2) - \hat{P}(x_s, a_s^1, a_s^2))^2$$
$$\leq \frac{1}{m}\sum_{i=1}^m (P(x_{s,i}, a_{s,i}^1, a_{s,i}^2) - \hat{P}(x_{s,i}, a_{s,i}^1, a_{s,i}^2))^2 + \frac{\log(2T|\mathcal{P}|/\delta)}{m},$$

which further implies that

$$\widetilde{\Gamma}_t^m(\lambda, \pi^1, \pi^2) \leq \sup_{P \in \mathcal{P}} \frac{|\mathbb{E}_x[P(x, \pi^1, \pi^2) - \hat{P}(x, \pi^1, \pi^2)]|}{\sqrt{\lambda - \frac{T\log(2T|\mathcal{P}|/\delta)}{m} + \frac{1}{2}\sum_{s=1}^{t-1}\mathbb{E}_{x_s \sim d_0, a_s^1 \sim \hat{\pi}_s^1, a_s^2 \sim \phi_s^2}(P(x_s, a_s^1, a_s^2) - \hat{P}(x_s, a_s^1, a_s^2))^2}}$$
$$\leq \sup_{P \in \mathcal{P}} \frac{|\mathbb{E}_x[P(x, \pi^1, \pi^2) - \hat{P}(x, \pi^1, \pi^2)]|}{\sqrt{\frac{1}{2}\lambda + \frac{1}{2}\sum_{s=1}^{t-1}\mathbb{E}_{x_s \sim d_0, a_s^1 \sim \hat{\pi}_s^1, a_s^2 \sim \phi_s^2}(P(x_s, a_s^1, a_s^2) - \hat{P}(x_s, a_s^1, a_s^2))^2}}$$
$$\leq \sup_{P \in \mathcal{P}} \frac{\sqrt{2} \cdot |\mathbb{E}_x[P(x, \pi^1, \pi^2) - \hat{P}(x, \pi^1, \pi^2)]|}{\sqrt{\lambda + \sum_{s=1}^{t-1}\mathbb{E}_{x_s \sim d_0, a_s^1 \sim \hat{\pi}_s^1, a_s^2 \sim \phi_s^2}(P(x_s, a_s^1, a_s^2) - \hat{P}(x_s, a_s^1, a_s^2))^2}}$$
$$\leq \sqrt{2} \cdot \Gamma_t(\lambda, \hat{\pi}_t^1, \hat{\pi}_t^2).$$

Here the second inequality is because $\lambda = \frac{2T\log(2T|\mathcal{P}|/\delta)}{m}$. Putting all together, we prove that with probability at least $1 - \delta$,

$$J(\pi_1^*, \pi_2^*) - J(\hat{\pi}_{t_0}^1, \dagger) \leq \mathbb{E}_{x \sim d_0}\left[3\beta\Gamma_{t_0}^m(\hat{\pi}_{t_0}^1, \hat{\pi}_{t_0}^2) - \eta^{-1}D_{\mathrm{KL}}(\hat{\pi}_t^2(\cdot|x)\|\widetilde{\pi}_t^2(\cdot|x))\right]$$
$$\leq 3\sqrt{2}\beta\Gamma_{t_0}(\lambda, \hat{\pi}_{t_0}^1, \hat{\pi}_{t_0}^2) - \eta^{-1}\mathbb{E}_x D_{\mathrm{KL}}(\hat{\pi}_t^2(\cdot|x)\|\widetilde{\pi}_t^2(\cdot|x))$$
$$\leq 3\sqrt{\frac{2T\log(2T|\mathcal{P}|/\delta)}{m}} - \eta^{-1}D_{\mathrm{KL}}(\hat{\pi}_t^2(\cdot|x)\|\widetilde{\pi}_t^2(\cdot|x)).$$

Setting $m = \frac{18T\log(2T|\mathcal{P}|/\delta)}{\epsilon^2}$ finishes the proof. $\qquad\square$

## D.1 Uncertainty for the Bradley-Terry Model

Recall that in Example 1, we suppose that the reward function can be embedded into a $d$-dimensional vector space $\{r(x, a) = \langle \theta, \phi(x, a) \rangle : \theta \in \mathbb{R}^d, \|\theta\| \leq B, \|\phi(x, a)\| \leq 1\}$. Then, if we define the covariance matrix as

$$\Sigma_t = \sum_{s=1}^{t-1}\mathbb{E}_{x \sim d_0, a^1 \sim \hat{\pi}_s^1, a^2 \sim \hat{\pi}_s^2}(\phi(x, a^1) - \phi(x, a^2))^\top(\phi(x, a^1) - \phi(x, a^2)) + \lambda(1 + e^B)^2 I.$$

By invoking the Lagrange's Mean Value Theorem, we have for any two parameters $\theta_1, \theta_2$,

$$\left|P_{\theta_1}(x, a^1, a^2) - P_{\theta_2}(x, a^1, a^2)\right| = \left|\frac{1}{1 + \exp(\theta_1^\top(\phi(x, a^2) - \phi(x, a^1)))} - \frac{1}{1 + \exp(\theta_2^\top(\phi(x, a^2) - \phi(x, a^1)))}\right|$$
$$\leq \left|(\theta_1 - \theta_2)^\top(\phi(x, a^2) - \phi(x, a^1))\right|,$$

and

$$\left|P_{\theta_1}(x, a^1, a^2) - P_{\theta_2}(x, a^1, a^2)\right| \geq \frac{1}{1 + e^B}\left|(\theta_1 - \theta_2)^\top(\phi(x, a^2) - \phi(x, a^1))\right|.$$

We use the short-hand notation $\phi(x, \pi) = \mathbb{E}_{a \sim \pi} \phi(x, a)$. Then the uncertainty can be bounded by

$$
\begin{aligned}
\Gamma_t(\lambda, \pi^1, \pi^2) &= \sup_\theta \frac{|\mathbb{E}_{x \sim d_0}[P_\theta(x, \pi^1, \pi^2) - P_{\hat\theta}(x, \pi^1, \pi^2)]|}{\sqrt{\lambda + \sum_{s=1}^{t-1} \mathbb{E}_{x_s \sim d_0, a_s^1 \sim \hat\pi_s^1, a_s^2 \sim \hat\pi_s^2}(P_\theta(x_s, a_s^1, a_s^2) - P_{\hat\theta}(x_s, a_s^1, a_s^2))^2}} \\
&\leq \sup_\theta \frac{\left|(\theta - \hat\theta)^\top \mathbb{E}_x[\phi(x, \pi^2) - \phi(x, \pi^1)]\right|}{\sqrt{\lambda + \sum_{s=1}^{t-1} \mathbb{E}_{x_s \sim d_0, a_s^1 \sim \hat\pi_s^1, a_s^2 \sim \hat\pi_s^2}(\frac{1}{1+e^B}|(\theta - \hat\theta)^\top(\phi(x, \pi^2) - \phi(x, \pi^1))|)^2}} \\
&\leq (1 + e^B) \sup_\theta \frac{\|\theta - \hat\theta\|_{\Sigma_t} \|\phi(x, \pi^1) - \phi(x, \pi^2)\|_{\Sigma_t^{-1}}}{\sqrt{(\theta - \hat\theta)^\top \Sigma_t (\theta - \hat\theta)}} \\
&= (1 + e^B)\|\phi(x, \pi^1) - \phi(x, \pi^2)\|_{\Sigma_t^{-1}}.
\end{aligned}
$$

This uncertainty bonus is consistent with that of the reward-based RLHF framework up to some multiplicative factor of regularization parameter [72] and the boundness parameter.

## D.2 Guarantee for Enhancer

---
**Algorithm 4** Optimistic Equilibrium Learning from Human Feedback with Enhancer Version 2
---

1: **Input:** Preference space $\mathcal{P}$, policy class $\Pi$, parameter $\lambda > 0$.
2: **for** t=1,…,T **do**
3:     Exploitation with the main agent: compute the MLE $\hat P_t$ with $\ell_{\mathcal{D}_{1:t-1}}$ defined in (7)
4:     Compute Nash equilibrium by calling the Oracle 2:

$$
\hat\pi_t^1 = \operatorname*{argmax}_{\pi^1 \in \Pi} \min_{\pi^2 \in \Pi} \mathbb{E}_{x \sim d_0, a^1 \sim \pi^1, a^2 \sim \pi^2}\left[\hat P_t(x, a^1, a^2) + \eta^{-1} \log \frac{\pi_0(a^1|x)}{\pi^1(a^1|x)} - \eta^{-1} \log \frac{\pi_0(a^2|x)}{\pi^2(a^2|x)}\right],
$$

5:     Exploration with the enhancer: construct bonus

$$
\widetilde\Gamma_t^m(\lambda, \pi^1, \pi^2) := \sup_{P \in \mathcal{P}} \frac{|\mathbb{E}_{x \sim d_0}[P(x, \pi^1, \pi^2) - \hat P_t(x, \pi^1, \pi^2)]|}{\sqrt{\lambda + \frac{1}{m}\sum_{s=1}^{t-1}\sum_{j=1}^{m}(P(x_{s,j}, a_{s,j}^1, a_{s,j}^2) - \hat P_t(x_{s,j}, a_{s,j}^1, a_{s,j}^2))^2}}. \quad (22)
$$

6:     Construct a version space for the policy

$$
\Pi_t = \{\pi \in \Pi : \eta^{-1}\mathbb{E}_x D_{\mathrm{KL}}(\pi(\cdot|x), \hat\pi^1(\cdot|x)) \leq \beta(\widetilde\Gamma_t^m(\lambda, \hat\pi^1, \pi) + \widetilde\Gamma_t^m(\lambda, \hat\pi^1, \hat\pi^1))\}.
$$

7:     Compute enhancer to maximize the uncertainty:

$$
\pi_t^2 = \operatorname*{argmax}_{\pi^2 \in \Pi_t} \widetilde\Gamma_t^m(\lambda, \hat\pi_t^1, \pi^2).
$$

8:     Collect $\mathcal{D}_t = \{(x_i, a_i^1, a_i^2, y_i)\}_{i=1}^m$ by $a_i^1 \sim \hat\pi_t^1(\cdot|x_i)$, $a_i^2 \sim \hat\pi_t^2(\cdot|x_i)$ and $y_i \sim \mathrm{Ber}\big(\mathbb{P}(a_i^1 \succ a_i^2 | x, a_i^1, a_i^2)\big)$;
9: **end for**
10: **Output:** the best policy in $(\pi_{1:T}^1)$ by a validation set.

---

**Lemma 2.** *Under Algorithm 4, given the policy of the main agent $\hat\pi_t^1$, we consider the version space with $\beta^2 = \log(2T|\mathcal{P}|/\delta)/m$:*

$$
\Pi_t = \{\pi \in \Pi : \eta^{-1}\mathbb{E}_x D_{\mathrm{KL}}(\pi(\cdot|x), \hat\pi^1(\cdot|x)) \leq \beta(\widetilde\Gamma_t^m(\lambda, \hat\pi^1, \pi) + \widetilde\Gamma_t^m(\lambda, \hat\pi^1, \hat\pi^1))\}.
$$

*Then, with probability at least $1 - \delta$, we know that $\operatorname{argmin}_{\pi'} J(\hat\pi_t^1, \pi') \in \Pi_t$ for all $t \in [T]$.*

*Proof.* First, since

$$
\hat\pi_t^1 = \operatorname*{argmax}_\pi \mathbb{E}_{a \sim \pi(\cdot|x)}\left[(1 - \hat P_t(x, \hat\pi_t^1, a)) - \eta D_{\mathrm{KL}}(\pi(\cdot|x)\|\pi_0(\cdot|x))\right],
$$

by using Lemma 9, we have for any policy $\pi \in \Pi$,

$$
\mathbb{E}_{x \sim d_0}\left[\mathbb{E}_\pi[1 - \hat P_t(x, \hat\pi_t^1, a)] - \mathbb{E}_{\hat\pi_t^1}[1 - \hat P_t(x, \hat\pi_t^1, a] + \eta D_{\mathrm{KL}}(\hat\pi_t^1(\cdot|x)\|\pi_0(\cdot|x)) - \eta D_{\mathrm{KL}}(\pi(\cdot|x)\|\pi_0(\cdot|x))\right]
$$
$$
= -\eta \mathbb{E}_{x \sim d_0} D_{\mathrm{KL}}(\pi(\cdot|x)\|\hat\pi_t^1(\cdot|x)),
$$

which implies that with $\pi = \widetilde{\pi}_t^2$,

$$\mathbb{E}_{x \sim d_0}\Big[ - \hat{P}_t(x, \hat{\pi}_t^1, \widetilde{\pi}_t^2) + \hat{P}_t(x, \hat{\pi}_t^1, \hat{\pi}_t^1) + \eta D_{\mathrm{KL}}(\hat{\pi}_t^1(\cdot|x)\|\pi_0(\cdot|x)) - \eta D_{\mathrm{KL}}(\widetilde{\pi}_t^2(\cdot|x)\|\pi_0(\cdot|x))\Big]$$
$$= - \eta \mathbb{E}_{x \sim d_0} D_{\mathrm{KL}}(\widetilde{\pi}_t^2(\cdot|x)\|\hat{\pi}_t^1(\cdot|x)). \tag{23}$$

Additionally, by the definition that

$$\widetilde{\pi}_t^2 = \operatorname*{argmin}_{\pi'} J(\hat{\pi}_t^1, \pi') = \operatorname*{argmin}_{\pi'} \mathbb{E}_x\big[P^*(x, \hat{\pi}_t^1, \pi') + \eta^{-1} D_{\mathrm{KL}}(\pi'(\cdot|x)\|\pi_0(\cdot|x))\big],$$

we have

$$\mathbb{E}_x\big[P^*(x, \hat{\pi}_t^1, \widetilde{\pi}_t^2) + \eta^{-1} D_{\mathrm{KL}}(\widetilde{\pi}_t^2(\cdot|x)\|\pi_0(\cdot|x))\big] \le \mathbb{E}_x\big[P^*(x, \hat{\pi}_t^1, \hat{\pi}_t^1) + \eta^{-1} D_{\mathrm{KL}}(\hat{\pi}_t^1(\cdot|x)\|\pi_0(\cdot|x))\big],$$

which implies that

$$0 \le \mathbb{E}_x\big[P^*(x, \hat{\pi}_t^1, \hat{\pi}_t^1) - P^*(x, \hat{\pi}_t^1, \widetilde{\pi}_t^2) + \eta^{-1} D_{\mathrm{KL}}(\hat{\pi}_t^1(\cdot|x)\|\pi_0(\cdot|x)) - \eta^{-1} D_{\mathrm{KL}}(\widetilde{\pi}_t^2(\cdot|x)\|\pi_0(\cdot|x))\big]$$
$$= \mathbb{E}_x\big[P^*(x, \hat{\pi}_t^1, \hat{\pi}_t^1) - \hat{P}_t(x, \hat{\pi}_t^1, \hat{\pi}_t^1) - (P^*(x, \hat{\pi}_t^1, \widetilde{\pi}_t^2) - \hat{P}_t(x, \hat{\pi}_t^1, \widetilde{\pi}_t^2))$$
$$\qquad + \hat{P}_t(x, \hat{\pi}_t^1, \hat{\pi}_t^1) - \hat{P}_t(x, \hat{\pi}_t^1, \widetilde{\pi}_t^2) + \eta^{-1} D_{\mathrm{KL}}(\hat{\pi}_t^1(\cdot|x)\|\pi_0(\cdot|x)) - \eta^{-1} D_{\mathrm{KL}}(\widetilde{\pi}_t^2(\cdot|x)\|\pi_0(\cdot|x))\big]$$
$$\le \beta(\widetilde{\Gamma}_t^m(\lambda, \hat{\pi}_t^1, \hat{\pi}_t^1) + \widetilde{\Gamma}_t^m(\lambda, \hat{\pi}_t^1, \widetilde{\pi}_t^2)) - \eta^{-1} \mathbb{E}_x D_{\mathrm{KL}}(\widetilde{\pi}_t^2(\cdot|x)\|\hat{\pi}_t^1(\cdot|x)),$$

where the last inequality uses (21) and (23) since $\hat{\pi}_1^t$ is the Nash equilibrium of $\hat{J}_t$. Therefore, we conclude the proof. $\qquad\square$

**Lemma 3.** *Under the same setting as Theorem 2, if we further assume that there exists a constant $B > 0$ such that for any $\pi \in \Pi$, $|\log(\pi(a|x)/\pi_0(a|x))| \le B$, and set $m = \frac{TB^4 \log(2T|\mathcal{P}|/\delta)}{\epsilon^2}$, we have with probability at least $1 - \delta$,*

$$J(\dagger, \hat{\pi}_{t_0}^2) - J(\pi^*, \pi^*) \le O(\epsilon^{1/2} - \eta^{-1} D_{\mathrm{KL}}(\hat{\pi}_{t_0}^2(\cdot|x)\|\widetilde{\pi}_{t_0}^2(\cdot|x))).$$

*Proof.* Under the condition of Theorem 2, we have

$$J(\hat{\pi}_t^1, \dagger) - J(\hat{\pi}_t^2, \dagger) \le \min_{\pi'} J(\hat{\pi}_t^1, \pi') - \min_{\pi'} J(\hat{\pi}_t^2 - \hat{\pi}_t^1, \pi') - \min_{\pi'} J(\hat{\pi}_t^1, \pi')$$
$$\le \min_{\pi'} \mathbb{E}_x\Big| \int (\hat{\pi}_t^1 - \hat{\pi}_t^2)(a|x) \cdot \pi'(a'|x) \cdot J(x, a, a') \mathrm{d}(a, a')\Big|$$
$$\le (B + 1)\mathbb{E}_x\|\hat{\pi}_t^1(\cdot|x) - \hat{\pi}_t^2(\cdot|x)\|_1$$
$$\le (B + 1)\sqrt{D_{\mathrm{KL}}(\hat{\pi}_t^1(\cdot|x)\|\hat{\pi}_t^2(\cdot|x))}$$
$$\le (B + 1)\sqrt{\eta\beta(\widetilde{\Gamma}_t^m(\lambda, \hat{\pi}_t^1, \hat{\pi}_t^2) + \widetilde{\Gamma}_t^m(\lambda, \hat{\pi}_t^1, \hat{\pi}_t^1))}$$
$$\le (B + 1)\sqrt{2\eta\beta\widetilde{\Gamma}_t^m(\lambda, \hat{\pi}_t^1, \hat{\pi}_t^2)},$$

where the second last inequality invokes Lemma 2, and the last inequality holds due to $\hat{\pi}_t^1 \in \Pi_t$ and $\hat{\pi}_t^2 = \operatorname{argmax}_{\pi \in \Pi_t} \widetilde{\Gamma}_t^m(\lambda, \hat{\pi}_t^1, \pi)$. Hence, at time $t_0$ in Theorem 2, we deduce that the suboptimality for the min-player is

$$J(\dagger, \hat{\pi}_{t_0}^2) - J(\pi^*, \pi^*) = J(\dagger, \hat{\pi}_{t_0}^2) - J(\dagger, \hat{\pi}_{t_0}^1) + J(\dagger, \hat{\pi}_{t_0}^1) - J(\pi^*, \pi^*)$$
$$= J(\hat{\pi}_{t_0}^1, \dagger) - J(\hat{\pi}_1^2, \dagger) + J(\pi^*, \pi^*) - J(\hat{\pi}_{t_0}^1, \dagger)$$
$$\le (B + 1)\sqrt{2\eta\beta\Gamma_{t_0}^m(\hat{\pi}_{t_0}^1, \hat{\pi}_{t_0}^2)} + 3\sqrt{\frac{2T\log(2T|\mathcal{P}|/\delta)}{m}} - \eta^{-1} D_{\mathrm{KL}}(\hat{\pi}_{t_0}^2(\cdot|x)\|\widetilde{\pi}_{t_0}^2(\cdot|x))$$
$$\le \mathcal{O}\Big( B\Big(\frac{T\log(2T|\mathcal{P}|/\delta)}{m}\Big)^{1/4} + \sqrt{\frac{T\log(2T|\mathcal{P}|/\delta)}{m}} - \eta^{-1} D_{\mathrm{KL}}(\hat{\pi}_{t_0}^2(\cdot|x)\|\widetilde{\pi}_{t_0}^2(\cdot|x))\Big)$$

Setting $m = \frac{TB^4 \log(2T|\mathcal{P}|/\delta)}{\epsilon^2}$, we get

$$J(\dagger, \hat{\pi}_{t_0}^2) - J(\pi^*, \pi^*) \le O(\epsilon^{1/2} - \eta^{-1} D_{\mathrm{KL}}(\hat{\pi}_{t_0}^2(\cdot|x)\|\widetilde{\pi}_{t_0}^2(\cdot|x))).$$

$\qquad\square$

# E Technical Lemmas

## E.1 Auxiliary Lemmas and Proofs

**Lemma 4.** *For $\max_{\pi^1 \in \Pi} \min_{\pi^2 \in \Pi} J(\pi^1, \pi^2)$, there exists a unique Nash equilibrium $(\pi_*^1, \pi_*^2)$ and it holds that $\pi_*^1 = \pi_*^2$.*

*Proof.* The existence and uniqueness of the Nash equilibrium are proved in Proposition 1 in [46]. We proceed to use the uniqueness of the Nash equilibrium and contradiction to prove the lemma. Suppose $\pi_*^1 \neq \pi_*^2$, since $\pi_*^1$ is the best response to $\pi_*^2$ for the max-player, for any $\pi \in \Pi$, we have

$$\mathbb{E}_{x \sim d_0} \left[ P(x, \pi, \pi_*^2) - \eta^{-1} D_{\mathrm{KL}}(\pi(\cdot|x) \| \pi_0(\cdot|x)) \right] \leq \mathbb{E}_{x \sim d_0} \left[ P(x, \pi_*^1, \pi_*^2) - \eta^{-1} D_{\mathrm{KL}}(\pi_*^1(\cdot|x) \| \pi_0(\cdot|x)) \right]. \tag{24}$$

Similarly, since $\pi_*^2$ is the best response to $\pi_*^1$ for the min-player, for any $\pi \in \Pi$, we have

$$\mathbb{E}_{x \sim d_0} \left[ P(x, \pi_*^1, \pi) - \eta^{-1} D_{\mathrm{KL}}(\pi(\cdot|x) \| \pi_0(\cdot|x)) \right] \geq \mathbb{E}_{x \sim d_0} \left[ P(x, \pi_*^1, \pi_*^2) - \eta^{-1} D_{\mathrm{KL}}(\pi_*^2(\cdot|x) \| \pi_0(\cdot|x)) \right]. \tag{25}$$

Then, we prove that $(\pi_*^2, \pi_*^1)$ is also the Nash equilibrium. Since $P(x, \pi^1, \pi^2) = 1 - P(x, \pi^2, \pi^1)$ for any $\pi^1$ and $\pi^2$, then (25) implies that for any $\pi \in \Pi$,

$$\mathbb{E}_{x \sim d_0} \left[ P(x, \pi, \pi_*^1) - \eta^{-1} D_{\mathrm{KL}}(\pi(\cdot|x) \| \pi_0(\cdot|x)) \right] \leq \mathbb{E}_{x \sim d_0} \left[ P(x, \pi_*^2, \pi_*^1) - \eta^{-1} D_{\mathrm{KL}}(\pi_*^2(\cdot|x) \| \pi_0(\cdot|x)) \right].$$

This demonstrates that $\pi_*^2$ is the best response to $\pi_*^1$ for the max-player. Similarly, (24) implies that for any $\pi \in \Pi$,

$$\mathbb{E}_{x \sim d_0} \left[ P(x, \pi_*^2, \pi) - \eta^{-1} D_{\mathrm{KL}}(\pi(\cdot|x) \| \pi_0(\cdot|x)) \right] \geq \mathbb{E}_{x \sim d_0} \left[ P(x, \pi_*^2, \pi_*^1) - \eta^{-1} D_{\mathrm{KL}}(\pi_*^1(\cdot|x) \| \pi_0(\cdot|x)) \right].$$

This demonstrates that $\pi_*^1$ is the best response to $\pi_*^2$ for the min-player. Hence, $(\pi_*^2, \pi_*^1)$ is another Nash equilibrium, contradicting with the uniqueness. Therefore, we have $\pi_*^1 = \pi_*^2$. $\qquad\square$

**Lemma 5.** *If $(\pi_*^1, \pi_*^2)$ is the Nash equilibrium of $\max_{\pi^1 \in \Pi} \min_{\pi^2 \in \Pi} J(\pi^1, \pi^2)$, then, we have*

$$(\pi_*^1(\cdot|x), \pi_*^2(\cdot|x)) = \operatorname*{argmax}_{\pi^1 \in \Pi} \operatorname*{argmin}_{\pi^2 \in \Pi} J(x, \pi^1, \pi^2)$$

*Proof.* According to Proposition 1 in [46], $(\pi_*^1, \pi_*^2)$ is the unique Nash Equilibrium of $\max_{\pi^1} \min_{\pi^2} J(\pi^1, \pi^2)$. According to the definition of the saddle point, it suffices to prove that for any $x \sim d_0$,

$$\pi_*^1(\cdot|x) = \operatorname*{argmax}_{\pi^1} J(x, \pi^1, \pi_*^2).$$

We know that

$$\begin{aligned} \pi_*^1 &= \operatorname*{argmax}_{\pi^1 \in \Pi} J(\pi^1, \pi_*^2) \\ &= \operatorname*{argmax}_{\pi^1 \in \Pi} \mathbb{E}_{x \sim d_0} \mathbb{E}_{a^1 \sim \pi^1, a^2 \sim \pi_*^2} [P^*(x, a^1, a^2) - \eta^{-1} D_{\mathrm{KL}}(\pi^1(\cdot|x) \| \pi_0(\cdot|x))]. \end{aligned}$$

Assume that there exists a $x_0$ such that

$$\pi_*^1(\cdot|x_0) \neq \widetilde{\pi}^1(\cdot|x_0) = \operatorname*{argmax}_{\pi^1 \in \Pi} \mathbb{E}_{a^1 \sim \pi^1, a^2 \sim \pi_*^2} [P^*(x, a^1, a^2) - \eta^{-1} D_{\mathrm{KL}}(\pi^1(\cdot|x) \| \pi_0(\cdot|x))].$$

Then we can construct a $\widetilde{\pi}_*^1 \in \Pi$ such that

$$\widetilde{\pi}_*^1(\cdot|x) = \pi_*^1(\cdot|x), \text{ for } x \neq x_0, \quad \widetilde{\pi}_*^1(\cdot|x_0) = \widetilde{\pi}^1(\cdot|x_0),$$

which contradicts the definition of $\pi_*^1$. Because of the symmetry of the two players, we also get

$$\pi_*^2(\cdot|x) = \operatorname*{argmin}_{\pi^2 \in \Pi} \mathbb{E}_{a^1 \sim \pi_*^1, a^2 \sim \pi^2} [P^*(x, a^1, a^2) + \eta^{-1} D_{\mathrm{KL}}(\pi^2(\cdot|x) \| \pi_0(\cdot|x))].$$

$\qquad\square$

## E.2 Other Lemmas

**Lemma 6** (Martingale Exponential Inequalities)**.** *Consider a sequence of random functions* $\xi_1(\mathcal{Z}_1), \cdots, \xi_t(\mathcal{Z}_t), \ldots$ *with respect to filtration* $\{\mathcal{F}_t\}$*. We have for any* $\delta \in (0, 1)$ *and* $\lambda > 0$:

$$\mathbb{P}\Big[\exists n > 0 : -\sum_{i=1}^{n} \xi_i \geq \frac{\log(1/\delta)}{\lambda} + \frac{1}{\lambda} \sum_{i=1}^{n} \log \mathbb{E}_{Z_i^{(y)}} \exp(-\lambda \xi_i)\Big] \leq \delta,$$

*where* $Z_t = (Z_t^{(x)}, Z_t^{(y)})$ *and* $\mathcal{Z}_t = (Z_1, \ldots, Z_t)$.

*Proof.* See Theorem 13.2 of Zhang [83] for a detailed proof. □

**Lemma 7** (Sion's minimax theorem)**.** *Let* $X$ *be a compact convex subset of a linear topological space and* $Y$ *a convex subset of a linear topological space. If* $f : X \times Y \to \mathbb{R}$ *satisfies*

- *for any fixed* $x \in X$, $f(x, \cdot)$ *is upper semicontinuous and quasi-concave on* $Y$;

- *for any fixed* $y \in Y$, $f(\cdot, y)$ *is lower semicontinuous and quasi-convex on* $X$,

*then we have*

$$\min_x \sup_y f(x, y) = \sup_y \min_x f(x, y).$$

**Lemma 8** (Multiplicative Chernoff Bounds)**.** *Assume that* $X \in [0, 1]$ *with* $\mathbb{E}X = \mu$*. Then for all* $\epsilon > 0$,

$$\mathbb{P}\Big(\bar{X}_n \geq (1 + \epsilon)\mu\Big) \leq \exp\Big[\frac{-2n\mu\epsilon^2}{2 + \epsilon}\Big]$$

$$\mathbb{P}\Big(\bar{X}_n \leq (1 - \epsilon)\mu\Big) \leq \exp\Big[\frac{-2n\mu\epsilon^2}{2}\Big].$$

*Moreover, for* $t > 0$, *we have*

$$\mathbb{P}\Big(\bar{X}_n \geq \mu + \sqrt{\frac{2\mu t}{n}} + \frac{t}{3n}\Big) \leq \exp(-t).$$

*Proof.* Refer to the proof of Corollary 2.18 in Zhang [83]. □

**Lemma 9** (Policy optimization error)**.** *For any two policies* $\pi, \hat{\pi} \in \Pi$ *such that* $\text{support}(\pi) = \text{support}(\pi_0)$ *and*

$$\hat{\pi}(\cdot|x) = \operatorname*{argmax}_{\pi^1 \in \Pi} \mathbb{E}_{a \sim \pi^1(\cdot|x)}\Big[\hat{P}(x, a) + \eta^{-1} \log \frac{\pi_0(a|x)}{\pi^1(a|x)}\Big],$$

*Suppose that the KL divergences between them are finite and well defined. Then, we have*

$$\mathbb{E}_{x \sim d_0}\Big[\mathbb{E}_\pi[\hat{P}(x, a)] - \mathbb{E}_{\hat{\pi}}[\hat{P}(x, a)] + \eta^{-1} D_{\mathrm{KL}}(\hat{\pi}(\cdot|x)\|\pi_0(\cdot|x)) - \eta^{-1} D_{\mathrm{KL}}(\pi(\cdot|x)\|\pi_0(\cdot|x))\Big]$$
$$= -\eta^{-1} \mathbb{E}_{x \sim d_0} D_{\mathrm{KL}}(\pi(\cdot|x)\|\hat{\pi}(\cdot|x)).$$

*Proof.* See the proof in Lemma 2.4 of Xiong et al. [72]. □

