# OpenReview forum: "Online Iterative Reinforcement Learning from Human Feedback with General Preference Model"
_NeurIPS.cc/2024/Conference — NeurIPS 2024 poster_

### Official Review · Reviewer_ZfnB · 2024-07-08

**Soundness:** 3
**Presentation:** 3
**Contribution:** 3
**Rating:** 7
**Confidence:** 3

**Summary:**

This paper explores RLHF with a general preference model. Specifically, the authors formulate a learning objective aimed at identifying a policy consistently preferred by the KL-regularized preference oracle. Furthermore, the authors propose a sample-efficient training algorithm applicable to both online and offline learning scenarios. The empirical studies conducted validate the effectiveness of the proposed framework.

**Strengths:**

1.	This paper delves into RLHF under a general preference oracle capable of capturing non-transitive preferences.

2.	The authors introduce an efficient algorithm applicable to both offline and online settings.

3.	The work is theoretically sound, and the authors also offer practical implementations.

4.	Experimental results demonstrate the effectiveness of the proposed method.

**Weaknesses:**

1.	It would be beneficial if the authors could include examples of the prompts used for training the preference model, as well as responses generated by different methods. This addition would offer readers a clearer understanding of the practical application and comparative effectiveness of the proposed approach.

2.	The reproducibility of the proposed method is not fully clear. Providing a detailed description of the entire framework or algorithm would greatly assist in understanding and replicating the research findings.

**Questions:**

1.	Is Assumption 1 considered mild in real-world applications?

2.	How does the preference model outperform the Bradley-Terry (BT) model, given that their training objectives in Equations 2 and 7 appear to be similar?

3.	Why does the proposed method outperform the baseline in the offline setting despite suffering from a severe overoptimization problem?

**Limitations:**

Yes

---

> ### Author Rebuttal · Authors · 2024-08-06
>
> Thanks for your positive feedback and constructive suggestions! Our responses are as follows.
>
> **Weakness 1** Include examples of the prompts used for training the preference model and responses generated by different methods.
>
> Sure. Some example prompts are provided as below
>
> ```markdown
> What is the ideal temperature to serve a crisp Chardonnay?
>
> is there xor method in vb?
>
> Define cryptography? Why we need cryptography?
>
> how to use viewdid load in swiftui macos xcode 14
> ```
>
> We also provide some example responses as below.
>
> ```markdown
> What are the names of some famous actors that started their careers on Broadway?
>
> DPO:
> 1. Hugh Jackman
> 2. Meryl Streep
> 3. Nathan Lane
> 4. Angela Lansbury
> 5. Bernadette Peters
> 6. Liza Minnelli
> 7. Audra McDonald
> 8. Philip Seymour Hoffman
> 9. Laurence Olivier
> 10. Kelsey Grammer
> 11. Richard Gere
> 12. Liev Schreiber
> 13. Mark Ruffalo
> 14. Julie Andrews
> 15. James Earl Jones
> 16. Whoopi Goldberg
> 17. Christopher Plummer
> 18. Robert Redford
> 19. Angela Bassett
> 20. Helen Mirren.
>
> IPO:
> Some famous actors that started their careers on Broadway include:
>
> 1. Hugh Jackman
> 2. Meryl Streep
> 3. Viola Davis
> 4. Denzel Washington
> 5. Mark Ruffalo
> 6. Matthew Broderick
> 7. Jennifer Holliday
> 8. Audra McDonald
> 9. Nathan Lane
> 10. Bernadette Peters
> 11. James Earl Jones
> 12. Angela Lansbury
> 13. Blythe Danner
> 14. Richard Dreyfuss
> 15. Cicely Tyson
>
>
> Ours:
> Here are some famous actors who started their careers on Broadway:
>
> 1. Meryl Streep - Before becoming one of the most acclaimed actors in Hollywood, Meryl Streep began her career on Broadway in the early 1970s. She made her debut in the play "The Play About the Baby" in 1971 and went on to star in several other productions, including "Trelawny of the 'Wells'" (1975), "Henry IV, Part 1" (1976), and "Sunday in the Park with George" (1984).
>
> 2. Hugh Jackman - Hugh Jackman made his Broadway debut in the musical "Oklahoma!" in 1990, playing the role of Curly. He went on to star in several other Broadway productions, including "Carousel" (1994), "The Iceman Cometh" (1999), and "The Boy from Oz" (2003), for which he won a Tony Award for Best Actor in a Musical.
>
> 3. Angela Lansbury - Angela Lansbury began her career on Broadway in the 1940s, starring in classic productions such as "Mrs. Warren's Profession" (1947), "The Picture of Dorian Gray" (1948), and "The Three Sisters" (1956). She won her first Tony Award for Best Actress in a Musical in 1966 for her performance in "Mame" and went on to win five more Tonys throughout her career.
>
> 4. Audrey Hepburn - Audrey Hepburn made her Broadway debut in the play "The Philadelphia Story" in 1956, playing the role of Tracy Lord. She went on to star in several other productions, including "Ondine" (1954) and "Tea and Sympathy" (1953), before transitioning to film.
> ```
> We omit some content for our answer because of the limit of characters in our response.
> It is clear that our proposed algorithm can enhance the helpfulness and engagement of the responses.
>
> **Weakness 2** The reproducibility of the proposed method is not fully clear.
>
> **We will make the description more detailed and release codes in the revised version.** We have 15K prompts for each iteration and perform 2 iterations for our algorithms. For each iteration, we have 2 epochs to train the policy model. For the exploration enhancer, we use the rejection sampling. The learning rate is 2e-7 with batch size 128.
>
> **Question 1** Is Assumption 1 mild in real-world applications?
>
> Yes. This assumption is quite standard in the RL literature. In practice, from Lines 122-123, we can allow an infinite function class as long as it has a finite covering number. Besides, when the capacity of the large language model is sufficiently large, it is very likely to contain the true model $P^*$.
>
> **Question 2** How does the preference model outperform BT model, given that their training objectives in (2), (7) appear to be similar?
>
> We want to clarify that the training objectives in Equations 2 and 7 are not similar, despite both maximizing an MLE objective. In (7), we directly learn the general preference $P(x,a^1,a^2)$ without assuming any specific reward structure, such as the BT model. This results in a preference function class with a much larger capacity. In contrast, (2) characterizes each response action pair by the reward $R(x,a)$ and assumes that the preference between two responses follows the BT model. The BT model imposes strong assumptions, which may not fully capture complex human preferences, such as intransitivity (see Lines 44-50).
>
> Overall, our training objective learns a more general preference function class, while Equation 2 only considers a preference function following the BT model. Therefore, our approach applies to more general cases.
>
> **Question 3** Why does the proposed method outperform the baseline in the offline setting despite suffering from a severe overoptimization problem?
>
> We reiterate that our algorithm should be less likely to have the overoptimization problem due to the better exploration and preference signal. Given the same data sample pairs, offline datasets only provide signals from the offline distribution, while our algorithm has better exploration strategy (equations (11, 12)) so that the overoptimization problem should be less severe.

---

> > ### Comment · Reviewer_ZfnB · 2024-08-11
> >
> > Thank you for your response. My concerns have been addressed. Accordingly, I've increased my score.

---

> > > ### Author Response · Authors · 2024-08-11
> > >
> > > Thank you for your reply and increasing your score! We are happy to see that our response addressed your concerns. Thanks again for your valuable feedback.

---

### Official Review · Reviewer_rwGu · 2024-07-10

**Soundness:** 3
**Presentation:** 2
**Contribution:** 2
**Rating:** 6
**Confidence:** 4

**Summary:**

This paper introduces a two-play game approach to the RLHF problems. Both offline and online algorithms are studied, including a combination of pessimism in offline learning and exploration/exploitation in the online setting. The authors conducted experiments to verify the effectiveness of their algorithm.
The paper is mostly well-written and clear, yet the experimental verification is not strong enough to support the paper's claims.

**Strengths:**

The paper is well-written, and I enjoyed reading it. The arguments are well supported by literature, and the authors are clearly knowledgeable and did well in presenting their ideas.

**Weaknesses:**

Mainly on novelty and soundness of empirical study.

**Questions:**

Could the authors elaborate more on the experiment details? From the current write-up, it is relatively hard for readers to capture how the training process is in either online or offline settings. Specifically,
1) how the reward models are trained? In Appendix A.2 the authors mentioned the bradley-terry RM and the oracle RM, is the oracle RM trained through next token prediction and with max length = 1 such that the probability of A is the score?
2) with those reward models, what is the exact policy optimization method used in the experiments (this is especially unclear for the offline setting).
3) Could the authors explain what are the motivations for the special choices in Section 5. Are there other alternatives?

The empirical results are impressive, however, I have some concerns regarding the fairness in comparisons:
1) the authors choose to compare mainly against offline methods, this seems not supportive enough for the proposed algorithm: could the authors also compare to other online RLHF algorithms? Only through this set of experiments will we be able to draw conclusions on the effectiveness of the proposed method --- rather than the superiority of online approaches over offline.
2) I reckon the implementation and reproduction of existing algorithms is not easy --- even only changing the dataset. Nonetheless, I would be keen to see some experiments regarding on how much effort would be needed in getting a well-performing algorithm. In other words, would it be easier to perform hyper-param sweeping for the proposed method than IPO / DPO (and their online alternatives)?

Ablation studies are required to draw conclusions of each element in the authors' empirical designs. For instance, the effectiveness of the pessimism objective, the necessity of each agent in the optimization problem (i.e., the min-player and its exploration.)

Regarding the novelty, could the author contrast the following literature (most of those are cited), and distinguish their contribution? I use [ - ] to denote their main message that is related to this paper:

[online RLHF is better than offline] https://arxiv.org/abs/2402.04792
[Nash learning from direct preference] https://arxiv.org/abs/2312.00886
[general preference optimization] https://arxiv.org/pdf/2402.05749
[self-play in RLHF] https://arxiv.org/abs/2401.01335

**Limitations:**

Please refer to the question section. My current score is mainly based on the soundness of the experiments.

---

> ### Author Rebuttal · Authors · 2024-08-06
>
> Thanks for your great efforts in reviewing our paper and constructive comments!
>
> **Question 1** Experiment details:
> 1. how reward models are trained? Is oracle RM trained through next token prediction and with max length = 1 such that probability of A is the score?
>
> We will add more details to improve the readability. For BT model, we remove the last layer of the LLaMA3-8B-it model and add a linear layer to construct the reward model, and train it by maximizing the log-likelihood on the preference dataset. For the oracle, yes, the training is conducted by next-token prediction. The problem is formulated as an instruction-following task, where we mask the prompt [CONTEXT] {x} [RESPONSE A] {a1} [RESPONSE B] {a2} and only compute the loss on the predicted token A or B. When inferencing, we re-normalize the probability on the two tokens and switch the positions of the two input responses to mitigate position bias.
>
> 2. Exact policy optimization method used in experiments (especially for the offline setting)
>
> For the online setting (Lines 234-251), in each iteration, we first learn the main policy by optimizing the self-play ipo loss in equation (13). Then, we learn the enhancer by using rejection sampling to choose the most uncertain policy with respect to the main policy. For the offline setting, when the offline dataset has ideally good coverage (uniform coverage over all policies), we can learn the policy via self-play ipo. However, such coverage is hard to guarantee, so we focus on the online algorithm and validate in Table 3 that the online algorithm performs much better than offline ones.
>
> 3. Motivations for the special choices in Section 5
>
> Our current choice matches best with our theoretical insights. The main player in Algorithm 2 is Nash equilibrium from oracle 2. Thus, practically, we apply self-play IPO to approximate the oracle. For the enhancer, we apply rejection sampling because equation (12) indicates that we need to maximize the uncertainty w.r.t. $\hat{\pi}_t^1$ within the confidence set. Since it is unknown how to compute the uncertainty for LLM, we practically use $\hat{\pi}_t^1$ to randomly generate $n$ samples, which is regarded as the confidence set. Then, we choose the best response in the confidence set, which aligns with the intuition of making the main agent and the enhancer more diverse. This process is the rejection sampling.
>
> **Question 2** Some concerns regarding the fairness in comparisons for empirical results.
> We answer this question from different aspects.
> - **Fairness:** Our comparison with offline benchmarks is relatively fair since our online algorithm does NOT require more preference label queries than the offline version. Specifically, we used 30K prompts divided into 2 iterations.
> - **Focus on Theory:** This paper primarily addresses theoretical aspects and proposes a framework to handle general preference. We newly conducted an iterative DPO algorithm (win rate: 14.37) compared to our algorithm's 17.67 win rate. While our algorithm outperforms the DPO, exploring the advantages of general preference over the BT model in practical scenarios (e.g., complex tasks like math and reasoning) has great potential for future work.
> - **Reproducibility：** The reproduction of our online algorithms is reliable and similar to other IPO/DPO algorithms. Some concurrent online DPO/IPO studies mainly emphasize empirical performance. Our paper provides robust theoretical support for these empirical findings. For hyperparam sweeping, our algorithm should also be similar to other online alternatives.
>
> **Question 3** Ablation studies.
>
> We will include more ablation studies in our paper. We illustrate some early results below
> | Baselines |  Full algorithm | Without enhancer exploration | Without Preference |
> | --- | --- | --- | --- |
> | AlpacaEval Win-Rate | 17.7 | 13.9 | 16.6 |
>
> We highlight that benefits from exploration align with other empirical studies, such as RLHF Workflow, who used rejection sampling as the exploration enhancer.
>
> **Question 4** Contrast following literature
>
> We will cite the rest and compare our work with them as follows. We denote these papers as:
> - [i] [online RLHF is better than offline]
> - [ii] [Nash learning from direct preference]
> - [iii] [general preference optimization]
> - [iv] [self-play in RLHF]
>
> Compared to these works focusing on planning (computing the optimal policy for a fixed model), our work considers the learning problem, which is on top of these planning algorithms, and further learns the optimal policy for the ground-truth model by interacting with the human and environment via exploration strategies.
>
> [i] shows that online and on-policy variants outperform offline counterparts, but the experiments are conducted with a fixed LLM for AI feedback. Therefore, no learning or exploration is considered in their work. Moreover, the paper is from an empirical side.
>
> [ii] also studies Nash learning and proposes a mirror-descent-based approach. Notably, the mixture of the reference policy and the current policy (see Eq. 3) in [ii] is challenging in practice due to the extremely large response space for LLM. Our algorithm employs a self-play strategy and does not require sampling from the geometric mixture, making it more practical. Further, [ii] focuses on the planning problem, whereas we address the learning problem and use exploration. Thus, the two works are complementary.
>
> [iii] proposes a unified framework with different loss functions and is also consistent with the results in [i] where DPO, IPO, and Slic perform similarly. They do not consider general preference oracle, learning, or exploration.
>
> [iv] studies self-play algorithm under BT model to approximate the distribution of offline dataset. However, in standard benchmarks like alpaca-eval, even the 7B model can beat GPT4 in many cases. In contrast, our work relies on an external general preference oracle and our goal is to learn the Nash policy under this oracle.

---

> > ### Comment · Reviewer_rwGu · 2024-08-11
> >
> > I appreciate the authors' detailed responses. I would encourage the authors to include those discussions in the revision of their paper.
> >
> > I've increased my score accordingly.

---

> > > ### Author Response · Authors · 2024-08-11
> > >
> > > Thank you for your reply and for increasing your rating!  We will integrate these discussions into the next revision.

---

### Official Review · Reviewer_kGMp · 2024-07-12

**Soundness:** 3
**Presentation:** 4
**Contribution:** 3
**Rating:** 7
**Confidence:** 4

**Summary:**

The paper considers the problem of RLHF under a general preference model, going beyond the reward-based Bradley-Terry model. In particular, they cast the problem as a KL regularized minimax game between two LLMs, and show that their framework is strictly more general than reward-based RL. They propose algorithms for both offline and iterative online learning under this framework, with guarantees. Finally, they run experiments comparing their work to existing RLHF algorithms like DPO and IPO.

**Strengths:**

1. One of the biggest strengths of this paper is its exposition and transparency. While certain nuances might be missed, it is clear that the authors have tried to convey their reasoning behind every conscious choice, and elaborate on the reasoning behind many folklore choices too.
2. The idea of deriving algorithms under a general preference model, while not radical, is both natural and formalizes the common technique of using larger LLMs to get preferential feedback for training smaller LLMs.
3. Going from a theoretically efficient but intractable algorithm to their tractable version requires novel thought, since the translation between the two here isn't as straightforward as it is in general.
4. I appreciate that they also tested on an OOD prompt set, despite this being a theory-first paper.

**Weaknesses:**

1. The paper should also have tested using $\log(p/(1-p))$ as a target, since that corresponds to the DPO target and an empirical comparison is also made to DPO. Their current target corresponds to the IPO target, as the paper implicitly recognises.
2. The theoretical version of the algorithm is quite standard and while this alone would not be a weakness, the practical version of the algorithm seems quite far from the theoretical version. Most strikingly, we use rejection sampling for the enhancer. I mentioned the latter as a strength due to the novel thought needed for designing the practical algorithm, but it is also a weakness because the theoretical guarantees do not signal much about the practical version due to this.
3. This is my main qualm: The iterative method queries the learned preference model adaptively, so it has a clear adaptive advantage over purely offline methods like DPO. The paper should also compare to iterative methods that use reward-based preference models, such as the one in [1].
4. There is also growing work in generalizing from standard reward based preference models to accommodate partial observability, which form an important special case of a fully general preference model. Works such as [2,3] are worth discussing and comparing to in related work.
5. This is not really a weakness, but some minor typos need to be fixed, such as a dangling [Proof] right after "Theorem 1" and "Theorem 2". Table 3's caption says offline DPO, but it should say offline IPO.

Refs:

1. RLHF Workflow: From Reward Modeling to Online RLHF. Dong et al, 2024.

2. When Your AIs Deceive You: Challenges of Partial Observability in Reinforcement Learning from Human Feedback. Lang et al, 2024.

3. A Theoretical Framework for Partially Observed Reward-States in RLHF. Kausik et al, 2024.

**Questions:**

1. Are their other options that the authors had while choosing a practical enhancer? I'm curious why they ended up choosing rejection sampling.
2. While the importance sampling ratio seems large in practice for the offline method, I'm wondering if there is a way to estimate the coverage coefficient too. This is a low priority question, but I am wondering if you have considered computing this to have a more compelling story.

**Limitations:**

Yes

---

> ### Author Rebuttal · Authors · 2024-08-06
>
> Thanks for your great efforts in reviewing our paper and thanks for recognizing our work!
>
> **Weakness 1**  The paper should also have tested using $\log(p/(1-p))$ as a target. Their current target corresponds to the IPO target.
>
> We use $P$ directly as a target since it is more straightforward to empirically approximate the oracle in Definition 2 by iterative IPO according to [1]. Note that our theoretical framework can also apply to $\log(p/(1-p))$. A concurrent work [2] uses $\log(p/(1-p))$ as the target and empirically approximates the Nash equilibrium by using some additional tricks, such as filtering preference pairs.
>
> [1] Daniele Calandriello, et al. Human alignment of large language models through online preference optimisation.
> [2] Rosset, Corby, et al. Direct nash optimization: Teaching language models to self-improve with general preferences.
>
> **Weakness 2** The practical version of the algorithm seems quite far from the theoretical version. Most strikingly, we use rejection sampling for the enhancer. I mentioned the latter as a strength due to the novel thought needed for designing the practical algorithm, but it is also a weakness because the theoretical guarantees do not signal much about the practical version due to this.
>
> We want to argue about the point that our theoretical and practical versions are disconnected since the online theoretical algorithm exactly provides insights for the empirical version. Specifically, equation (12) in the theory part illustrates that instead of making $\hat{\pi}_t^2$ the same as the main policy $\hat{\pi}_t^1$, we need to maximize the uncertainty with respect to $\hat{\pi}_t^1$ within the confidence set. Since it is unknown how to compute the uncertainty for LLM, in practice, we use $\hat{\pi}_t^1$ to randomly generate $n$ samples, which is regarded as the confidence set. Then, we choose the best response in the confidence set, which aligns with the intuition of making the main agent and the enhancer more diverse. This process exactly is the rejection sampling.
>
> **Weakness 3** The iterative method queries the learned preference model adaptively, so it has a clear adaptive advantage over purely offline methods like DPO. The paper should also compare to iterative methods that use reward-based preference models, such as the one in [1].
>
> Thanks for your question. First, to ensure fairness, we guaranteed that the number of the preference queries is less than the number of the offline datasets. We use 2 iterations (15K prompt for each) for our online methods, while offline datasets contain 60K paired responses. Second, we will also add comparison with iterative DPO methods under our computation and query number setting. In our setting, it performs slightly worse than our algorithm (14.37 win rate, compared with ours 17.67). In general, for chat optimization, our algorithm should be similar to or slightly better than iterative DPO since BT model seems to be a reasonable assumption in this case as indicated by the similar chat accuracies of BT reward and preference model.
>
> Besides, this paper mainly focuses on theory and proposes a framework to handle the general preference. To explore more advantages of general preference over BT-model in real practice, we need much work on complicated tasks, like math and reasoning, which should be future work.
>
> **Weakness 4** Works such as [2,3] that accommodate partial observability are worth discussing and comparing to in related work.
>
> Thanks for this point! We will add some discussions in the revision. We want to kindly remind that the models with partial observability and our general preference setting are two separate lines of work. Our setting considers a general preference function class when the BT-model does not hold, while the partial observability setting considers that the state features cannot be observed. The combination of the two lines of study would be interesting for future work.
>
> **Weakness 5** Some minor typos need to be fixed.
>
> Thanks for pointing out the typos. We will correct them in the revision.
>
> **Question 1** Are their other options that the authors had while choosing a practical enhancer? Why they ended up choosing rejection sampling.
>
> Of course, there are other options, but the rejection sampling aligns best with our theoretical insights. The reasons have been stated in Weakness 2.
>
> One reasonable explanation is that if the reward/preference model is well-calibrated, which means that the underlying ground-truth accuracy of judging aligns well with the confidence of the model, then, a large margin between the two responses means a higher accuracy. With rejection sampling, the learning signal is more accurate and is also more consistent.
>
> **Question 2** If there is a way to estimate the coverage coefficient too. This is a low priority question, but I am wondering if you have considered computing this to have a more compelling story.
>
> Thanks for this point! The importance sampling ratio is an upper bound of the coverage coefficient and we present some case studies in the paper where it is usually large. An accurate estimation of the coverage coefficient itself can be challenging but our general impression is that we usually cannot expect to have a moderate coverage coefficient even for a single policy. The indirect evidence is when we personally tried to apply the offline algorithms in the literature to real-world RL applications, we eventually found that for almost all the cases, the resulting policy can and can only compete with the behavior policy.
>
> Therefore, the main applicable situation of offline learning is that we have a noisy expert who can frequently make mistakes and the offline learning algorithm can automatically adapt to the successful trajectory.
>
> We also remark that when we are considering the Markov game, a *unilateral concentration* is required, which can be even more challenging compared to the single-policy coverage.

---

> ### Comment · Reviewer_kGMp · 2024-08-12
> **Raising score**
>
> The authors have addressed my concerns on an already excellently written paper. I am raising my score to a 7. I believe that this paper fully deserves to be accepted.

---

> > ### Author Response · Authors · 2024-08-12
> >
> > Thank you for your reply and for raising the score! We are happy to see that our response has addressed your concerns. Thanks again for your constructive feedback.

---

### Official Review · Reviewer_Bkdx · 2024-07-13

**Soundness:** 2
**Presentation:** 3
**Contribution:** 2
**Rating:** 5
**Confidence:** 3

**Summary:**

The authors develop a theoretical framework based on a reverse-KL regularized minimax game and introduce sample-efficient algorithms suitable for both offline and online learning scenarios. Empirical results validate the proposed method, demonstrating its superior performance compared to traditional reward-based models across various tasks.

**Strengths:**

The paper tackles a critical issue in existing RLHF approaches, challenging the assumption that human preferences are transitive. This is significant given the evidence of intransitivity in human decision-making.
The paper is well-written and easy to follow.
Empirical results demonstrate the advantages of the proposed approach over baseline methods.

**Weaknesses:**

The empirical validation, while promising, is limited in scope. More extensive experiments, including comparisons with a broader range of state-of-the-art RLHF methods, would strengthen the paper.

**Questions:**

How can we maintain the consistency of the learning process in the absence of the transitive assumption? For instance, when preferences are conflictive.

**Limitations:**

NA.

---

> ### Author Rebuttal · Authors · 2024-08-06
>
> Thanks for your constructive comments! Our responses are as follows.
>
> **Weakness 1** The empirical validation, while promising, is limited in scope. More extensive experiments, including comparisons with a broader range of state-of-the-art RLHF methods, would strengthen the paper.
>
> Thank you for your question. We conducted large-scale experiments demonstrating that our algorithm performs comparably to iterative DPO in LLM chat benchmarks. We observed that the BT model is a reasonable assumption in this context, as indicated by the similar chat accuracies between the BT reward and preference models. However, to fully explore the advantages of a general preference model over the BT model in real-world applications, more work on complex tasks, such as mathematics and reasoning, is required. This will be the focus of our future research.
>
> **Question 1** How can we maintain the consistency of the learning process in the absence of the transitive assumption? For instance, when preferences are conflictive.
>
> Thanks for this point! As stated in Section 2, we directly optimize the general preference and do not make any assumptions about the general preference, such as transitivity. Our algorithm aims to learn the Nash equilibrium policy as defined in Definition 2. For the Nash policy, it is consistently preferred by the KL-regularized preference in the face of any competing policy (line 126). Therefore, the Nash policy inherently accounts for intransitivity among different responses.
>
> In our experiments, we use self-play IPO (Lines 234-240) to approximate the Nash equilibrium oracle. Intuitively, due to the symmetry of the preference $P$, the max-player $\pi_1^*$ equals the min-player $\pi_2^*$ when Nash equilibrium is achieved. Hence, instead of solving the min-max problem, we iteratively compute the best response to the last iteration.

---

### Author Rebuttal · Authors · 2024-08-06

Thank all the reviewers for the constructive comments. We appreciate that all reviewers provided positive feedback during the initial review round. We would like to highlight some key points below to clarify some confusing parts.

1. Novelty — Online algorithm with general preference oracle: we propose algorithms and provide rigorous analysis on the statistical complexity of the iterative RLHF algorithms under **general preference oracle**. Furthermore, due to the limit of exploration for the offline methods, we focus on the online iterative framework and propose a practical algorithm Online ELHF IPO. The advantage of Online ELHF IPO is validated by experiments.
2. Main focus — Theoretical framework and proof-of-concept experiments: This paper mainly focuses on theory and proposes a framework to handle the general preference. We provide controlled experiments to show that our proposed preference-based online algorithm demonstrates advantages over baselines.
3. Comparison with other concurrent online algorithms: From the theoretical perspective, our work is the first online algorithm to study the general preference. For empirical algorithms, there are some concurrent algorithms. Considering the experimental comparison with some concurrent SOTA online algorithms, our algorithm is at least comparably good than others in chat benchmarks, which is also indicated by the similar accuracy between BT & preference model. More significant empirical advantages of the proposed framework should be on reasoning-related benchmarks. However, due to the page limit, the large-scale experiments are out of the scope of this paper. Our paper can be regarded as both theoretical foundation and empirical proof of concept. More comprehensive empirical studies should be future work.

---

### Decision · Program_Chairs · 2024-09-25

**Decision:**

Accept (poster)

**Comment:**

This work makes an interesting contribution to the growing literature on online RLHF by considering a minimax game between two LLMs for RLHF under general preference oracle. All the reviewers agreed that the theoretical results of the paper are sound, and novel. Regarding the experimental part, some reviewers raised concerns that comparing the online RLHF method with the existing offline RLHF method might not be fair. However, the authors have addressed this concern in the follow-up response. Overall, I think this paper makes a useful contribution to the active literature on RLHF. It would be great if the authors could add the details and discussions regarding the experiments in the next version of the paper.